# The Anti-Oxidative, Anti-Inflammatory, Anti-Apoptotic, and Anti-Necroptotic Role of Zinc in COVID-19 and Sepsis

**DOI:** 10.3390/antiox12111942

**Published:** 2023-10-31

**Authors:** George Briassoulis, Panagiotis Briassoulis, Stavroula Ilia, Marianna Miliaraki, Efrossini Briassouli

**Affiliations:** 1Postgraduate Program “Emergency and Intensive Care in Children, Adolescents, and Young Adults”, School of Medicine, University of Crete, 71003 Heraklion, Greece; stavroula.ilia@uoc.gr; 2Second Department of Anesthesiology, Attikon University Hospital, School of Medicine, National and Kapodistrian University of Athens, 12462 Athens, Greece; briaspan@med.uoa.gr; 3Paediatric Intensive Care Unit, University Hospital, School of Medicine, University of Crete, 71110 Heraklion, Greece; med1p1130027@med.uoc.gr; 4Infectious Diseases Department “MAKKA”, First Department of Paediatrics, “Aghia Sophia” Children’s Hospital, School of Medicine, National and Kapodistrian University of Athens, 11527 Athens, Greece; ggbriass@med.uoc.gr

**Keywords:** micronutrient, zinc, bioavailability, antioxidant, anti-inflammatory, A20, programmed cell death, apoptosis, necroptosis, supplementation, sepsis, COVID-19

## Abstract

Zinc is a structural component of proteins, functions as a catalytic co-factor in DNA synthesis and transcription of hundreds of enzymes, and has a regulatory role in protein–DNA interactions of zinc-finger proteins. For many years, zinc has been acknowledged for its anti-oxidative and anti-inflammatory functions. Furthermore, zinc is a potent inhibitor of caspases-3, -7, and -8, modulating the caspase-controlled apoptosis and necroptosis. In recent years, the immunomodulatory role of zinc in sepsis and COVID-19 has been investigated. Both sepsis and COVID-19 are related to various regulated cell death (RCD) pathways, including apoptosis and necroptosis. Lack of zinc may have a negative effect on many immune functions, such as oxidative burst, cytokine production, chemotaxis, degranulation, phagocytosis, and RCD. While plasma zinc concentrations decline swiftly during both sepsis and COVID-19, this reduction is primarily attributed to a redistribution process associated with the inflammatory response. In this response, hepatic metallothionein production increases in reaction to cytokine release, which is linked to inflammation, and this protein effectively captures and stores zinc in the liver. Multiple regulatory mechanisms come into play, influencing the uptake of zinc, the binding of zinc to blood albumin and red blood cells, as well as the buffering and modulation of cytosolic zinc levels. Decreased zinc levels are associated with increasing severity of organ dysfunction, prolonged hospital stay and increased mortality in septic and COVID-19 patients. Results of recent studies focusing on these topics are summarized and discussed in this narrative review. Existing evidence currently does not support pharmacological zinc supplementation in patients with sepsis or COVID-19. Complementation and repletion should follow current guidelines for micronutrients in critically ill patients. Further research investigating the pharmacological mechanism of zinc in programmed cell death caused by invasive infections and its therapeutic potential in sepsis and COVID-19 could be worthwhile.

## 1. Introduction

Sepsis and septic shock pose significant healthcare challenges, impacting millions of individuals globally on an annual basis and resulting in the mortality of around 17–33% of those afflicted [1]. Sepsis can give rise to life-threatening organ dysfunction, brought about by an imbalanced immune reaction to pathogens [2], characterized by systemic inflammation and heightened oxidative stress. Critically ill patients with COVID-19, a public health emergency of international concern in recent years, can inherently be considered septic, as most of them meet the sepsis definition with acute respiratory distress syndrome (ARDS) [3]. COVID-19, however, is characterised by “silent hypoxemia”, alveolar thrombosis, protracted lung inflammation, and coagulopathy with highly elevated circulating fibrinogen and D-dimers [4]. Both in sepsis and COVID-19, apoptotic and necroptotic regulated cell death (RCD) modes are initiated and propagated by molecular mechanisms that exhibit a considerable degree of interconnectivity manifested with varying immunomodulatory profiles [5]. Multiple innate immune defence pathways intersect at the caspase-8 level, interweaving different RCD pathways, which create complex signalling networks that cross-guard each other against pathogens and prevent pathogens from inhibiting the function of extrinsic apoptosis and necroptosis [6]. Moreover, each type of RCD can manifest with an entire spectrum of features ranging from anti-inflammatory and tolerogenic to pro-inflammatory, immunogenic and immunomodulatory profiles, characterized by necrotic and apoptotic morphological features [5].

Apoptosis and necroptosis release many danger signals, such as damage-associated molecular patterns (DAMPs), inflammatory mediators, anti-inflammatory metabolites or molecules packaged in apoptotic extracellular vesicles, nucleosomal structures, shed receptors, and phosphatidylserine molecules, contributing to the inflammatory response and exacerbation of sepsis and COVID-19 [7]. In both sepsis and COVID-19, the severity of systemic inflammation, oxidative stress, and organ dysfunction are negatively associated with zinc concentrations [8]. Recently, an increasing number of studies addressed the zinc immunomodulatory role and its relation to various cell death pathways in sepsis and COVID-19. In this narrative review, we describe the protective role of zinc against oxidative stress and the programmed cell death-regulatory pathways in sepsis and COVID-19 and summarize the current knowledge obtained by clinical studies.

## 2. Materials and Methods

We searched the databases PubMed and Science Direct for eligible studies published up to June 2023. The keywords used in the search were “zinc”, “oxidative stress”, “antioxidant, “cell death”, “apoptosis”, “necroptosis”, “critical illness”, “sepsis”, and “Coronavirus disease 2019 (COVID-19)”. Boolean operators “AND” and “OR” were used alone or as combined descriptors. Studies that presented relevant aspects of the role of zinc in the protection against sepsis or COVID-19 were included. Articles were excluded if they were duplicated in different databases. Two independent reviewers (SI and PB) screened titles and abstracts of 452 included articles. Full-text readings were carried out by two independent reviewers (MM and EB) and any disagreement between them was resolved through discussions with a third reviewer (GB). In the next step, we selected the full text for 135 articles and included 117 additional articles after screening the reference lists for relevant studies.

## 3. Zinc Functions

Zinc is an essential micronutrient needed to ensure the proper functioning of crucial biochemical and metabolic processes. The role of zinc in the body can be grouped into three general functional classes: (a) structural component in proteins; (b) catalytic co-factor in DNA synthesis and RNA transcription of hundreds of enzymes such as polymerases, carboxy peptidases, superoxide dismutase, and carbonic anhydrase [9]; and (c) regulatory in protein–DNA interactions of zinc-finger proteins, including many transcription factors [10], inhibitory of viral replication [11], and modulatory of inflammation [12] and oxidative stress [13]. Among them, the most extensively studied regulatory functions are antioxidant activity, the effect on inflammation, immune regulations, and regulated cell death.

### 3.1. Antioxidant Activity

Although the production of reactive oxygen species (ROS) aims to eradicate pathogens, excessive oxidative stress in sepsis leads to cell membrane lipid peroxidation, protein oxidation, mitochondrial damage, inflammation, and increased inducible nitric oxide synthetase (iNOS)-induced NO, apoptosis and necroptosis [14]. The main antioxidant proteins, including superoxide dismutase (SOD), glutathione, catalase, thioredoxin, and peroxiredoxin, regulate the redox balance in the mitochondria [15]. In a clinical study, elderly individuals who received zinc supplementation exhibited notably reduced rates of infections and lower levels of plasma oxidative stress markers when compared to those in the placebo group [16]. Hence, zinc could be of potential benefit as antioxidant therapy for redox balance restoration in sepsis and COVID-19 [17].

Zinc acts as an antioxidant by neutralizing free radicals directly by glutathione, affecting the expression of the rate-limiting enzyme of de novo synthesis glutathione, glutamate-cysteine ligase [18]. Zinc enhances the effectiveness of Cu/Zn SOD by serving as a structural component. This enzyme transforms superoxide (O_2_^−^) into the less harmful hydrogen peroxide (H_2_O_2_) and oxygen (O_2_), thereby reducing the harmful effects of reactive oxygen species (ROS) [19]. Metallothioneins are involved in the reduction of hydroxyl radicals, acting as cytoprotective agents and electrophilic scavengers, maintaining zinc-related cell homeostasis [20].

In murine sepsis, zinc was shown to activate the nuclear factor-erythroid 2-related factor 2 (Nrf2) transcription factor, which regulates glutathione peroxidase 2 (GPx2), glutamate-cysteine ligase catalytic subunit (GCLC), thioredoxin reductase 1, members of the glutathione-S-transferase (GST), heme-oxygenase-1 (HO-1), and SOD by binding to the anti-oxidant responsive cis-acting enhancer sequence elements [21]. Also, different forms of zinc supplementation (zinc methionine, zinc sulphate, and nano zinc oxide) improved serum and seminal plasma SOD and catalase, along with physiological parameters, of heat-stressed male rabbits [22]. Zinc was shown to increase metallothionein in the liver of endotoxin/ Zn^2+^-adequate diet rats, inhibit endotoxin lethality in Zn^2+^-treated animals, and protect against endotoxin-associated mediators [23].

The indirect antioxidant function of zinc is to compete with redox-active metals for negative charges in the lipid bilayer [24], protecting the cell membrane from the lipid oxidation [25], and interacting with thiol and sulfhydryl groups, preventing intramolecular disulfide formation [26]. A second mechanism consists of antagonizing metal-catalysed transition reactions [20] by exchanging redox-active metals, such as copper and iron, in certain binding sites attenuating cellular site-specific oxidative injury in the setting of severe infection. It is part of the antioxidant defence to stabilize cytosolic Zn/Cu superoxide dismutase, inhibiting the pro-oxidant enzyme nicotinamide adenine dinucleotide phosphate oxidase (NADPH)-Oxidase producing superoxide radicals (O_2_^−^) and inducible nitric oxide synthase (iNOS) [27], and inducing the expression of cysteine-rich metallothioneins [28].

Experimentally, anti-oxidant genes were noticeably improved with a dietary zinc nanoparticles diet and iNOS, heat shock protein-70, and DNA damage-inducible protein, were significantly upregulated in fish [29] and mice [30]. In rats, zinc oxide nanoparticles demonstrated antioxidant properties by effectively reducing serum cyclooxygenase-2 (COX-2) enzyme activity, decreasing tissue nuclear factor—kappa B (NF-κB) levels, and lowering blood hypoxia-inducible factors (HIF-1α) levels [31]. Supplementation of zinc-enriched probiotics of Wistar rats significantly enhanced glutathione content, glutathione-peroxidase and superoxide-dismutase activity, and decreased malondialdehyde content while the expression of glutathione peroxidase 1 (GPx1) and SOD1 genes significantly increased [32]. Also, exposure of magur catfish to zinc oxide nanoparticles, stimulated the NO production by hepatocytes because of induction of iNOS activity, higher expression of nos2 gene and iNOS protein [33].

### 3.2. Effect on Inflammation

Various pro-inflammatory transcription factors, such as NF-κB and activator protein-1 (AP-1), have the capability to trigger the generation of ROS, leading to the secretion of inflammatory cytokines. This, in turn, exacerbates oxidative stress [34]. Cellular transcriptional response to ROS is mediated mainly by activation of mitogen-activated protein (MAP) kinases, cysteines Cys65 and Cys93 in redox effector factor (Ref-1), the redox-dependent modification of transcription factors p53, activating transcription factor/cAMP-response, element–binding protein (ATF/CREB), hypoxia-inducible factor (HIF)-1α, and HIF-like factor [35,36]. Superoxide anions undergo oxidative phosphorylation with the oxidative systems consisting of xanthine oxidase, nicotinamide adenine dinucleotide phosphate (NADPH) oxidases, 5-lipoxygenase, and cyclooxygenase [37]. Strong experimental evidence shows that the proteasome partially processes phosphorylation of p105 and p100 precursor NF-κB proteins at S893 and S907 residues to produce NF-κB subunits p50 and p52. The p105 and p100 precursor proteins contain C-terminal ankyrin repeats conferring IκB-like function by sequestering interacting NF-κB subunits in the cytoplasm. The p50 NF-κB subunit is phosphorylated by protein kinase A (PKA) at S337, by checkpoint kinase 1 (Chk1) at 328 triggered by DNA damage, and by DNA-dependent protein kinase (DNA-PK) responded to TNF-α stimulus at S20 [38].

NF-κB is phosphorylated downstream in responding to toll-like receptors (TLRs), antigens, cytokines, growth factor receptors, oxidative stimuli, and viral products. In the cytoplasm, the phosphorylation of NF-κB at specific NH_2_-terminal serine residues degrades the inhibitory subunit inhibitors of κB (IκB), activating the IκB kinase (IKK) complex of IKKα and IKKβ. Phosphorylation of IκBα by the IKK complex triggers its recognition by beta-transducing repeats-containing proteins (β-TrCP), substrate recognition subunits for the SCFβ-TrCP E3 ubiquitin ligases, leading to polyubiquitination and proteasome degradation. The released NF-κB dimers then translocate to the nucleus, where they recognise and bind specific DNA sequences, termed κB sites, activating the transcription of many different target-proinflammatory genes [39]. The NF-κB signalling pathway regulates inflammatory responses through the IκB kinase (IKK) complex to promote cell survival. The activation of IKKα and IKKβ, induced by TNF stimulation and involving the ubiquitylation of receptor-interacting serine/threonine-protein kinase-1 (RIPK1) and NF-κB essential modifier (NEMO), plays a pivotal role in promoting cell survival through the NF-κB signalling pathway [40]. TNF receptor 1 (TNFR1) ligation activates the NF-κB signaling pathway acting as a cell death determinant because NF-κB transcription factors induce the expression of anti-apoptotic genes. Also, in the TGFβ-activated kinase-1 (TAK1) complex, TAB2 contains Lys63-linked ubiquitin-binding domains, interacts with RIPK1 and NEMO [41], and activates NF-κB signalling by initiating the IKK complex [42]. NF-κB activation increases the expression of the adhesion molecules E-selectin, VCAM-1, and intercellular adhesion molecule 1 (ICAM-1) and mediates the synthesis of cytokines and cyclooxygenase 2 (Cox-2) [39]. NF-κB has also been shown to function in concert with other transcription factors, such as AP-1 [43].

Upon zinc supplementation, zinc receptor metal regulatory transcription factor 1 (MTF-1) and other transcription factors responsible for cellular adaptation to oxidative stress enhance gene expression within the nucleus. This enhancement occurs by increasing DNA binding and recruiting various co-activators, thereby maintaining cellular homeostasis [44]. The release of zinc from metallothioneins activates NF-κB and influences its ability to transmit pro-inflammatory signals to the nucleus, regulating the expression of DNA for transcription factors involved in the inflammation process. Inflammatory states have shown a coordinated interplay among zinc, metallothioneins, MTF-1, and pro-inflammatory cytokines like IL-6 and TNF-α, orchestrated by the transcription factor NF-κB [44]. Zinc can attenuate the pro-inflammatory response through cell uptake by the zinc transporter protein ZIP14, also known as SLC39A14 (Solute Carrier Family 39 Member 14), which plays a crucial role in zinc homeostasis within the body. ZIP14 facilitates the movement of zinc from storage sites (liver) to immune cells at the site of infection in sepsis, by facilitating the redistribution of zinc to immune cells, supporting immune function, modulating inflammation, and aiding in antioxidant defence [45]. Current evidence indicates that zinc modulates NF-κB signalling at various levels, primarily by regulating the activity of proteins and enzymes involved in the NF-κB pathway. Some ways zinc can influence NF-κB signalling are the inhibition of IKK activation, enhancement of IκB stability, modulation of redox status, regulation of zinc finger proteins, influence on immune cell function, and epigenetic regulation [46]. Zinc can inhibit the activation of IKK, a kinase that phosphorylates IκB (inhibitor of NF-κB), leading to its degradation. This prevents the release and translocation of NF-κB to the nucleus [47]. Simultaneously, zinc enhances the stability of IκB, which sequesters NF-κB in the cytoplasm. The activation of the NF-κB heterodimer, composed of p65-p50 and IκBα, occurs through the phosphorylation of IκB by the IκB kinase (IKK) complex. This leads to the ubiquitination and subsequent proteasomal degradation of IκB [48]. Zinc, through the inhibition of cyclic nucleotide phosphodiesterase (PDE), causes an increase in cyclic guanosine monophosphate (cGMP). This rise in cGMP activates protein kinase A (PKA) and, by phosphorylation, inhibits the protein kinase Raf-1. Moreover, zinc suppresses the activation of IKKβ induced by LPS [20]. Also, imported by the zinc transporter ZIP8, cytosolic zinc induces NF-κB inhibition downstream from mitogen-activated protein kinases (MAPKs) by blocking the IKK complex [49]. Finally, zinc negatively regulates the NF-κB pathway by inducing the expression of zinc-finger protein A20, which is the main negative regulator of the NF-κB activation [48]. Cytosolic zinc, transported via the zinc transporter ZIP8, induces the inhibition of NF-κB downstream of MAPKs by interfering with the IKK complex [47]. This, in turn, affects its target genes, such as TNF-α and IL-1-beta (IL-1β) [50], as well as chemoattractant proteins like monocyte chemoattractant protein-1 (MCP-1), and intercellular adhesion molecule-1 (ICAM-1) [51]. Additionally, zinc can indirectly modulate the NF-κB pathway by affecting the activity of immune cells that produce cytokines and other molecules involved in NF-κB signalling [52]. Furthermore, zinc can play a role in epigenetic regulation by influencing DNA methylation and histone modifications, potentially affecting the expression of genes involved in the NF-κB signalling pathway [53].

Zinc exerts an influence on the cellular redox status, mitigating oxidative stress and countering the activation of NF-κB signalling [54]. Given its importance for the function of various zinc finger proteins, some of which can interact with NF-κB, zinc can impact the activity and localization of NF-κB. Zinc fingers are autonomously folded domains structured around a zinc ion [55] with roles in DNA recognition, RNA packaging, transcriptional activation, protein folding, assembly, and regulation of necroptosis [56]. Research indicates that zinc plays a regulatory role in NF-κB transcription, mediated by the zinc-finger protein TNF-α-induced protein 3 (TNFAIP3) and the receptor signalling pathway activated by peroxisome proliferator-activated receptor (PPAR) [57,58]. TNFAIP3, also known as A20, is a widely expressed cytoplasmic signalling protein known for its anti-inflammatory, NF-κB inhibitory, and anti-necroptotic properties. A20 comprehensively regulates ubiquitin-dependent signals and modulates the duration and intensity of signalling by various proteins involved in the NF-κB pathway [57,59,60]. In TNF receptor (TNFR)- and toll-like receptor (TLR)-initiated pathways, TNFAIP3 serves as the primary negative regulator of NF-κB activation, impacting endothelial cell adhesion molecules and oxidative stress biomarkers [51]. Studies have shown that A20 functions as a negative regulator, maintaining the balance in the strength and duration of NF-κB activation by deubiquitinating receptor-interacting serine/threonine-protein kinase 1 (RIPK1) and TNF receptor-associated factor 2 (TRAF2), the components of the TNFR1 signalling complex (Figure 1). Moreover, A20’s deubiquitylase (DUB) activity restricts TRAF6-mediated and RIPK2-mediated activation of NF-κB during TLR/IL-1R and NOD signalling, respectively. In vitro, zinc upregulates the gene expression of A20 and peroxisome proliferator-activated receptor alpha (PPAR-α), both of which are zinc finger proteins with anti-inflammatory properties [20].

### 3.3. Immune Regulation

In sepsis, DAMPs, such as plasma mitochondrial DNA (mtDNA), serve as drivers of an immune or inflammatory response [61]. The DAMPs are released during sepsis-induced immune cell death, which is linked with multiple organ dysfunction [62]. Increasing evidence has shown that circulating mtDNA can trigger innate immunity by activating the Toll-like receptor 9/NF-κB pathway [63] or the NOD, LRR, and pyrin domain-containing protein 3 (NLRP3) inflammasome [64]. Zinc, in conjunction with various micronutrients, plays a role in DNA metabolism, resulting in reduced DNA damage and lowered levels of mtDNA in the bloodstream [65]. Conversely, lower zinc concentrations have been correlated with higher levels of mtDNA in patients with HIV and those coinfected with HIV and hepatitis C [66]. In an animal model, the presence of extracellular mtDNA activated the NLRP3 inflammasome, initiating inflammation via the Toll-like receptor 9 (TLR-9), mitogen-activated protein kinase (MAPK), and NF-κB pathways [67]. When monocytes were stimulated with mtDNA in vitro, it led to the production of TNF-α. Moreover, elevated levels of mtDNA in the plasma were correlated with increased in vivo levels of pro-inflammatory cytokines, including TNF-α, IL-6, Regulated upon Activation, Normal T Cell Expressed and Presumably Secreted (RANTES), and IL-1 receptor antagonist (IL-1ra) [68]. In both in vitro and in vivo settings, mtDNA activated neutrophils by promoting the phosphorylation of MAPK and inducing migration and degranulation of human polymorphonuclear neutrophils. This activation of neutrophils was mediated through TLR-9 and formyl peptide receptor-1 (FPR1) [69]. This relationship suggests that DAMPs might contribute to the persistent and dysregulated inflammatory response, leading to processes like apoptosis and necroptosis [62] and exacerbating organ dysfunction during sepsis [70]. The use of a zinc-polysaccharide complex has the potential to mitigate mitochondrial damage and reduce the production of cytokines induced by lipopolysaccharides (LPS) by inhibiting the MAPK signalling pathway [71].

Zinc is crucial for the appropriate development and function of innate and adaptive immunity and the affected signalling cascades in sepsis [72]. Regarding innate immunity, intracellular zinc is critical for the extravasation of neutrophils to the site of the infection and the uptake and killing of microorganisms [73]. Moreover, A20′s capacity to interact with ubiquitin enzyme complexes is essential for fine-tuning ubiquitin-dependent innate immune signalling pathways, including those linked to TNFR1, TLRs, IL-1R, CD40, and NOD-like receptors (NLRs). Low blood zinc, which is often found during the early stages of sepsis because of inflammation, can have negative effects on phagocytosis, oxidative burst, degranulation, cytokine production, chemotaxis [73], NK cell function, and lytic activity [74]. Zinc deficiency, both in vivo and in vitro, has been shown to result in impaired granulocyte migration and chemotaxis in response to N-formyl-L-methionyl-L-leucyl-L-phenylalanine (fMLF), as well as a reduction in the chemoattractant properties of IL-8 for T cells [73]. Moreover, zinc deficiency affects the modulation of the inflammatory response by IL-1ra on the cell surface [75]. In individuals with zinc deficiency, there is a demonstrated decrease in the exocytosis of neutrophil granules and secretory vesicles of macrophage-1 antigen (Mac-1), which is a complement receptor-3 (CR3), along with CD66b, a signal transducer involved in the adhesive activity of CD11/CD18 [73]. This reduction in exocytosis negatively impacts critical immune functions such as phagocytosis, oxidative burst, and granule release [76]. This reduction in exocytosis negatively impacts critical immune functions such as phagocytosis, oxidative burst, and granule release [77]. Furthermore, zinc deficiency leads to a decrease in the production of thymulin, which is essential for the differentiation and functioning of T cells [78]. It also impairs the production of Th1 cytokines, including TNF-α, IL-2, and IFN-γ, both in cell culture models [79] and in vivo [80].

Regarding adaptive immunity, zinc restores lymphocyte production [81] and natural killer (NK) activity [82] and promotes autophagy in vitro by maintaining the proper structure and function of zinc-dependent lysosomal enzymes, including proteases, peptidases, phosphatases, nucleases, glycosidases, sulfatases, and lipases [83]. In TLR- and TNFR-triggered signalling pathways, A20 was shown to downregulate the expression of IL-1β, TNF-α, CRP [58] and inhibit T- and B-cell-induced NF-κB signalling, interacting with proteins that bind to ubiquitin and the zinc finger domain of Inhibitor of κB kinase (IKK) gamma (IKK-γ) [20]. Zinc signals mediated by the zinc transporter ZIP8, a Zrt-/Irt-like protein metal transporter with multiple natural substrates, influence the adaptive immune response during bacterial pneumonia, protecting lung epithelium [84,85].

Zinc is a cofactor for many enzymes in microorganisms and can directly affect microbial growth and function, affect their virulence, and toxin production and either guarantee or disrupt key processes for survival [86]. It has been recently shown that pathogens attacking peripheral blood mononuclear cells disrupt their immune function, impairing phagocytosis in sepsis [87]. These data suggest bacterial uptake and lysosomal degradation by monocytes or macrophages during sepsis and invasive infections are inhibited, leading to excessive lysosomal load with hid to zinc bacteria or viruses into dysfunctional monocytes or macrophages [88,89]. Furthermore, zinc overexposure shifts gut microbiota to antibiotic-resistance genes [90] by increasing the zinc available for bacteria [91], interrupting the barriers to nutritional immunity [92]. While zinc is essential in small amounts, excessive zinc can disrupt microbial cell membranes, interfere with essential cellular processes, and ultimately lead to microbial cell death. Thus, zinc sequestration by the human immune system facilitated by interleukin IL-6 helps to intoxicate engulfed pathogens and act cytoprotectively by neutralizing ROS and nitrogen species [93,94].

Stimulation of nucleotide-binding oligomerisation domain-2 (NOD2) leads to an increase in intracellular zinc levels, which, in turn, enhances autophagy and promotes the clearance of bacteria. The regulation of metallothionein genes in human monocyte-derived macrophages (MDMs) relies on the zinc-dependent transcription factor MTF-1 [95]. Knocking down MTF-1 does not impact the baseline bacterial clearance by MDMs. However, when there is continuous NOD2 stimulation, the increase in intracellular zinc, autophagy, and bacterial clearance is compromised in MDMs lacking MTF-1 [95]. MTF-1 is involved in the transcription of genes responsible for sequestering and transporting zinc within cells and regulating intracellular signalling pathways by activating metallothioneins [96]. MTF-1 moves between the cytoplasm and nucleus, with its export likely mediated by its interaction with chromosomal maintenance 1 (Crm1), a major mammalian export protein facilitating the transport of large macromolecules, including RNA and proteins, across the nuclear membrane to the cytoplasm (Figure 2). The addition of zinc or induction of autophagy restores bacterial clearance in MDMs after metallothionein knockdown. In addition, in MDMs, NOD2 works synergistically with the Pattern Recognition Receptors (PRRs) TLRs 5 and 9 to enhance the effects of metallothioneins [95]. In mice, the intestinal microbiota plays a role in regulating the expression of metallothioneins, zinc levels, autophagy, and bacterial clearance in intestinal macrophages [95].

Similarly, in interaction with the activating transcription factor 4 (ATF4), HIF-1 improves the immune activity of macrophages involved in sepsis [97]. It has been demonstrated that inhibiting glycolysis contributes to neutrophil immunosuppression during sepsis, regulated by the PI3K/Akt-HIF-1α-pathway-mediated downregulation of lactate dehydrogenase A (LDHA) [98]. SARS-CoV-2 ORF3a induces mitochondrial damage and the production of mitochondrial ROS, leading to increased HIF-1α expression, which subsequently facilitates SARS-CoV-2 infection and cytokine production [99]. The effects of HIF-1α on vasculature, metabolism, inflammation, immune response, and apoptosis contribute to lung injury in COVID-19 [100]. Zinc finger and bric-a-brac (BTB)-domain-containing protein 2 (ZBTB2) enhance the expression of specific HIF-1 target genes under hypoxia. HIF-1 binding to the consensus hypoxia-responsive element (HRE) sequence recruits ZBTB2 to the gene locus, increasing p300-mediated histone acetylation and enhancing gene expression under hypoxic conditions (Figure 2). In normoxia, prolyl hydroxylases (PHDs) hydroxylate HIF-1α on two proline residues in the oxygen-dependent degradation domain, triggering von Hippel–Lindau (pVHL)-mediated ubiquitination and proteasomal degradation. Concurrently, factor inhibiting HIF (FIH), an asparaginyl hydroxylase regulated similarly to PHDs in an oxygen-dependent manner, suppresses HIF-1′s transcriptional activity in normoxia by preventing co-activator recruitment. In contrast, hypoxia inhibits PHDs and stabilizes HIF-1α, allowing it to translocate into the nucleus and dimerize with constitutively expressed HIF-1β, forming an active HIF-1 complex that triggers the transcription of genes promoting glycolytic metabolism, angiogenesis, and cell survival [101]. Both in vitro and in vivo studies confirm that zinc promotes angiogenesis through the HIF-1α/VEGF-signalling pathway [102] (Figure 2).

### 3.4. Hormonal Effects

The pituitary gland boasts the most substantial zinc concentration, which augments the performance of pituitary hormones [103]. When there is a shortage of zinc, it leads to a deficiency in the secretion of pituitary growth hormone (GH), nerve growth factor (NGF), insulin-like growth factor 1 (IGF-1) [104], and hippocampal nerve growth factor 7S (NGF-7S) [105]. Importantly, COVID symptoms might also be due to hypothalamic–pituitary–adrenal (HPA) axis suppression induced by the virus and the immune system. The angiotensin receptor converting enzyme 2 (ACE2) expression in the zona fasciculata and zona reticularis of the adrenal cortex, injured by the identified in autopsies COVID-19 virus, could affect glucocorticoid synthesis [106]. ACE2 is expressed not only in the lower respiratory tract but also in the somatosensory cortex, rectal/orbital gyrus, temporal lobe, hypothalamus/thalamus, brainstem, and cerebellum. Direct viral infection of the neuronal cells in these regions occurs by the virus binding to the ACE2 receptor, resulting in demyelination and neurodegeneration and inducing a greater risk of long-term effects in some patients recovering from COVID-19 [107]. Zinc deficiency and genetic susceptibility also contribute to neurodegeneration [108]. Thus, the elevated zinc deficiency gene for the odorant metabolizing enzyme UDP-glucuronosyltransferase (UGT) has been linked to COVID-induced anosmia [109].

While COVID-19 can induce stress-related metabolic disturbances and zinc deficiency can impact the proper functioning of insulin receptors on cell surfaces, the exact interactions between insulin resistance and zinc in the context of COVID-19 are not fully understood. In vivo, zinc improved the endothelium-dependent vasorelaxation, reversed the reduction of guanosine 5′-triphosphate cyclohydrolase 1 and tetrahydrobiopterin, and suppressed the elevation of ROS in streptozotocin-induced diabetic mice [110]. Finally, epigenetic changes on the male and female reproductive organs secondary to COVID-19 exposure are not well documented, although reports indicating adverse effects of COVID-19 on male and female gametogenesis are emerging [111]. SARS-CoV-2 induces cytokine storm, increases ROS in the gametogenic cells, and depletes intracellular zinc, potentially resulting in oocyte and sperm damage. Furthermore, zinc deficiency mediating ROS overproduction potentiates oocyte and sperm damage, aggravating the effects of COVID-19 [111].

## 4. Regulated Cell Death

Regulated cell death, known as programmed cell death (PCD), refers to the form of cell death regulated by a variety of biomolecules involving molecularly defined signalling pathways and effector mechanisms, which is distinctive from the uncontrolled biological process of accidental cell death (ACD) [112]. Single or mixed types of RCD may occur in response to acute oxidative stress associated with infectious or non-infectious human diseases [113]. ROS-induced lipid peroxidation and DNA damage may cause genomic instability, and structural damage, resulting in cell rupture, stimulating cell death pathways, and triggering inflammation. Zinc deficiency promotes cell death by leading to oxidative stress and ROS production [114]. According to the guidelines established by the Nomenclature Committee on Cell Death, distinct molecular mechanisms give rise to distinctive characteristics within major cell death subroutines [5]. Furthermore, oxidative stress is not just associated with apoptotic-regulated cell death (apoptosis) but is also implicated in non-apoptotic varieties of regulated cell death, including necroptosis, ferroptosis, and pyroptosis [113].

### 4.1. Apoptosis

Apoptosis is an important pathway of programmed cell death in sepsis [115,116]. Linked to apoptosis are caspases, which are cysteine proteases that cleave their substrates on the aspartate C-terminal side, leading to apoptotic vesicle formation [117,118]. Intrinsic apoptosis is triggered by various disturbances in the microenvironment, such as ROS, endoplasmic reticulum (ER) stress, DNA damage, alterations in microtubules, or defects in mitosis. The crucial event in intrinsic apoptosis is the irreversible and widespread permeabilization of the mitochondrial outer membrane (MOMP), which is regulated by pro-apoptotic and anti-apoptotic members of the Bcl-2 protein family. This process activates caspase-9 and caspase-3. On the other hand, extrinsic apoptosis is a form of regulated cell death initiated by changes in the extracellular microenvironment. Extrinsic apoptosis is initiated either by death receptors (such as the Fas cell surface death receptor and TNF receptor superfamily member 1A), leading to the formation of a dynamic multiprotein complex at the intracellular domain of the receptor, or by dependence receptors (like netrin 1 receptors and neurotrophin receptor neurotrophic receptor tyrosine kinase 3). Activation of dependence receptors occurs when the levels of their specific ligands fall below a certain threshold, and this activation subsequently leads to the activation of caspase-8 and caspase-3 [5]. In the context of sepsis, which is induced by an increased load of ROS, inflammatory biomolecules, and mitochondrial damage, the total concentrations of caspases-3, -8, and -9 (including pro-caspases, active forms, and cleaved forms) are elevated. This suggests that both the extrinsic and intrinsic pathways are involved in sepsis-induced apoptosis [119].

Zinc, being a potent inhibitor of caspases-3, -7, and -8, modulates the caspase-controlled inflammatory processes [120]. Zinc uptake into cells through the zinc importer protein ZIP10 is essential for the B cell survival [121] and apoptotic signalling cascades of the Bcl/Bax family and DNA fragmentation [122]. Importantly, zinc inhibits the mediators within the apoptotic pathway of caspases-3, -6 and -8 (Figure 1) in a dose-dependent manner [123]. The provision of zinc through supplementation enhances the expression of zinc transporters, specifically ZnT-4 and ZnT-7, responsible for shuttling zinc from the cytosol to the mitochondria [124]. Notably, ZnT-7 appears to be associated with the activation of tyrosine kinase proteins. When activated, these tyrosine kinase proteins inhibit apoptosis by phosphorylating pro-apoptotic proteins [125]. Zinc deficiency in these organelles appears to be associated with oxidative stress pathways of the Bcl-2 family with subsequent activation of caspase-3 [124], triggering apoptosis in B and T cells [126]. Experimental studies have demonstrated the implication of zinc in apoptosis in various organic tissues in sepsis and COVID-19 [127,128].

Regarding sepsis, experimental studies have suggested a potential implication of zinc in apoptosis in various tissues during sepsis. A deficiency of zinc led to imbalanced ratios of Th1- and Th2-cells, an elevated rate of apoptosis in immature T cells, and consequently, an overall reduction in the total number of T cells [129]. Among pre-T cells (CD4+CD8+) isolated from zinc-deficient mice, apoptosis was intensified by 50–300%. This resulted in a 38% reduction in the pre-T cell population within the thymus, along with an 80% decrease in the total thymic cell count [130]. Furthermore, zinc deficiency resulted in diminished lytic activity of natural killer cells and a lower percentage of precursor cells for cytolytic T cells [131]. Zinc chelation exacerbated LPS-induced apoptosis in pulmonary endothelium and reversed the protective effect of NO donors on LPS-induced apoptosis [127]. Elevated zinc contents in septic lung and kidney tissues suggested that apoptosis of erythrocytes (eryptosis) is characterized by phosphatidylserine expression on the outer membrane. Eryptosis is also enhanced, whereas erythrocyte adhesion to endothelial cells was increased after zinc stimulation in vitro [132]. Also, the number of PBMCs with active caspase-3/7 and the number of apoptotic cells decreased in a dose-dependent way in the presence of zinc aspartate [133].

Intestine epithelial cells are apoptotic and shed in sepsis. Paneth cells maintain gut mucosal integrity and homeostasis. In an experimental study, septic mice exhibited significantly fewer Paneth cells in the crypt, with disorganized granules or without granules [134]. In murine sepsis, increased ileal expression of TLR4 promoted the depletion of Paneth cells, and reduced lysozyme and defensin alpha 5 (DEF-5) expression exacerbated intestinal damage, and increased mortality rate. It has been suggested that TLR4 mediates the hyperactivation of ER stress, leading to Paneth cell loss and dysfunction during intestinal barrier impairment of sepsis [134].

COVID-19 infection is associated with an excessive release of high mobility group box-1 (HMGB1) in human lungs, triggering multiple pathways of cell death [135]. COVID-19 infection is associated with an excessive release of high mobility group box-1 (HMGB1) in human lungs, triggering multiple pathways of cell death [136,137], potentially including the induction of a senescent phenotype [128]. The gastrointestinal system serves as another active site for the replication of SARS-CoV-2, particularly in mature enterocytes expressing the ACE2 viral receptor and TMPRSS4 protease. Symptoms related to the gastrointestinal tract may manifest prior to the onset of respiratory symptoms and are observed in 5–80% of COVID-19 patients [138]. Within Paneth cells, the nucleotide-binding oligomerization domain 2 (NOD2), a recognition receptor that detects conserved motifs of MDP bacterial peptidoglycan in the cytosol, is expressed. Upon interaction with MDP, NOD2 triggers the production of host defence peptides (HDPs), cytokines, and chemokines, stimulating an immune response from both epithelial and immune cells [138]. Additionally, Paneth cells generate antimicrobial proteins such as C-type lectin REG3γ, α-defensins, β-defensins, cathelicidins, and lysozyme in response to infection [139], which may indirectly impact COVID-19 by influencing the microbiome [140]. Notably, in mice lacking ZnT2, a zinc transporter primarily located in Paneth cell granules [96], abnormal Paneth cell granules were observed, resulting in reduced bacterial killing by crypt secretions compared to those of wild-type secretions. This suggests that the absence of zinc impaired Paneth cell secretion [141].

In vitro, depletion of intracellular zinc in Paneth cells, which contain a large amount of zinc in their granules [142], impaired nuclear translocation of the p65 subunit of NF-κB and up-regulation of cIAP2 mRNA, compromised tight junctions’ integrity and intestinal permeability, and increased intestinal cells’ apoptosis [143]. Deleting SLC39A7, encoding ZIP7 that controls zinc transport from the endoplasmic reticulum (ER) to cytosol, leads to substantial damage to the intestine, degeneration of post-mitotic Paneth cells with destructive granules, collapsed ER structure, and apoptosis in intestinal inflammation [144], whereas upregulation of *ZIP7* expression is associated with regeneration of Paneth cells [145]. Since apoptotic cells emit signals that can induce apoptosis in neighbouring cells, the deletion of ZIP7 might lead to extensive apoptosis in crypt cells, with the apoptotic remnants being engulfed by nearby viable stem cells. [55]. High zinc diet (2500 mg/kg) in weaned piglets infected with transmissible gastroenteritis virus, down-regulated interferon (*IFN*)*-**α* and *ZIP4*, up-regulated metallothionein-1, *ZnT1*, and *ZnT5*, prevented villus atrophy of jejunal epithelium, and decreased caspase-3-mediated apoptosis [146].

### 4.2. Necroptosis

Necroptosis is a form of cell death that combines features of both apoptosis and necrosis. Morphologically, it resembles necrosis, featuring organelle and cell swelling, plasma membrane rupture, and the release of intracellular components, all without the chromatin condensation typically seen in apoptosis [5]. It can be governed by underlying genetic programs [147]. Necroptosis is controlled by receptor-interacting kinases 3 (RIPK3), the kinase activity of RIPK1, and their substrate, mixed lineage kinase domain-like protein (MLKL), which promotes its oligomerization and activation. The execution of necroptosis is orchestrated by complex IIb, also known as the necrosome, which lies downstream of TNF-R1 activation. In cells undergoing necroptosis, the rapid permeabilization of membranes, driven by the executioner protein MLKL, leads to the release of intracellular contents. Notably, caspase-8 serves as a negative regulator of necroptosis (Figure 1).

Both in sepsis and COVID-19, necroptosis seems to have a pivotal role in the development of tissue injury. In sepsis, RIPK1 has dual functions and not only can regulate necroptosis in endothelial cells by the TNF and TLR pathways leading to sepsis-induced organ failure [148], but also activates NF-κB via ubiquitination at different sites, induces inflammation states, stabilizes cell lysosome function, induces autophagic activity, and maintains intracellular homeostasis [149]. In murine sepsis, RIPK1 has been shown to induce the production of GM-CSF, IL-6, IL-8, and TNF under LPS stimulation, implying that RIPK1 has a detrimental impact on sepsis [150,151]. Necrostatin-1 (Nec-1), an antagonist of RIPK1, has the capacity to hinder TNF-induced sepsis by reducing the levels of circulating proinflammatory cytokines and DAMPs, reversing the increased vascular and intestinal permeability, and inhibiting the activation of the coagulation cascade in vascular endothelial cells [152,153,154]. In animal experiments involving sepsis models, Nec-1 demonstrated the ability to mitigate systemic and pulmonary inflammation [155]. It also lessened jejunal injury and improved digestive and barrier functions by inhibiting RIPK1 [156]. However, it is worth noting that cells lacking RIPK1 exhibited various lysosomal abnormalities, including compromised lysosomal activity, acidity, and the expression of lysosome-related genes [157]. While targeting the kinase activity of RIPK1 could potentially limit the rapid escalation of inflammation, such an intervention did not consistently meet treatment expectations when sepsis was complicated and intracellular homeostasis was dynamically changing [158,159]. Consequently, the equilibrium between RIPK1′s pro-survival functions and the activation of its pro-death kinase activity may influence the clinical outcome during different stages of sepsis [160].

COVID-19 induces cytokine storm, releasing DAMPs such as the mobility group box 1 (HMGB1), leading to necroptosis programmed cell death [161], organ damage related to viral burden, and mortality [162]. In COVID-19, the necroptosis pathways of MLKL-, fas-associated death domain (FADD)-, and apoptosis-associated speck-like protein containing a caspase recruitment domain (ASC)-related genes are likely to be involved in lymphopenia and might potentially predict the COVID-19 outcome [163]. Importantly, MLKL is a biomarker of necroptosis in the hearts of COVID-19 patients and a therapeutic target for major adverse cardiac events [164]. It has been recently shown that alveolar type II (ATII) pneumocytes succumb to TNF-α-induced necroptosis and new PANoptotic forms of programmed cell death in COVID-19 pneumonia [165]. Accordingly, the clinical transformation of necroptosis inhibitors might be a novel therapeutic option for the treatment of COVID-19 [166].

Zinc and zinc-binding proteins participate in the process of necroptosis positively and negatively. Zinc, zinc finger protein 91 (ZFP91), and zinc-finger transcription factor Sp1, facilitate activation of RIPK3-dependent necroptosis or induces mitochondrial ROS, promoting TNF-mediated RIPK3-independent necroptosis [167]. Specifically, ZFP91 initiates necroptosis by inducing TNFR1 ligation via the phosphorylation or oligomerization of RIPK1-RIPK3-MLKL complex, enhancing necrosome formation, and promoting RIPK1-RIPK3-MLKL signalling pathway activation [167] (Figure 1). Along with *TNFR1*, *RIPK1*, *RIPK3*, and *MLKL,* the zinc transporter ZIP7 is a significant necroptosis effector protein, contributing to the inflammatory phenotypes observed in TNIP1 (TNFAIP3 interacting protein 1, ABIN-1)-deficient mice [168]. TNIP1 influences intracellular signalling through its binding partner ubiquitin-editing enzyme [169,170].

In contrast to ZIP7, zinc finger protein A20 antagonizes necroptosis [171] and inhibits macrophage necroptosis [172]. A20 inhibits necroptotic RCD by preventing excessive formation of the RIPK1-RIPK3 necroptotic complex through inhibition of kinase RIPK3 ubiquitination at the Lys5 residue [173]. Zinc transporter ZIP7 affects all aspects of TNFR1 signalling, controlling zinc efflux from the endoplasmic reticulum (ER) to the cytoplasm on the necroptosis pathway activation [174]. High zinc concentration in mesenchymal stem cells inhibits folding factors in the ER, including protein disulfide isomerases (PDIs) [174] and antagonizes TNF-induced necroptosis in enterocytes via modulating intestinal microbiota [175].

While TNIP1 deficiency renders cells more susceptible to TNF-α-induced necroptosis by impacting the activation of RIPK1 [176], TNFα-induced TNIP1 is recruited to the TNF receptor signalling complex, where it interacts with A20 to deactivate RIPK1, thus inhibiting necroptosis [177]. Overexpression of TNIP1 prevents the phosphorylation of RIPK1, RIPK3 [178], and MLKL, thereby regulating membrane rupture and controlling TNF-α-induced necroptosis [168]. Zinc finger protein 91 (ZFP91) stabilizes both the RIPK1 and RIPK3 proteins, promoting their interaction and facilitating the process of necroptosis. It contributes to the formation of complex IIb, which consists of caspase-8, FADD, MLKL, RIPK1, and RIPK3. This formation can be induced by inhibiting the caspase-8-mediated cleavage of RIPK [179]. Within complex IIb, caspase-8 cleaves RIPK1, resulting in pro-apoptotic caspase activation and apoptosis [180], whereas the inhibition of caspase-8 rescues RIPK1, promoting necroptosis by facilitating the formation of the necrosome [181]. In parallel, ZFP91 induces de-ubiquitination of RIPK1 and triggers the activation of the necrosome, initiating necroptosis (Figure 1).

## 5. Zinc Bioavailability

The bioavailability of zinc in critically ill patients is frequently affected by various mechanisms [182,183]: (1) Zinc bioavailability can be influenced by various factors, including the specific formulations of zinc used, the route of administration, and the zinc status of the individuals participating in the trials; (2) Several factors can impact the absorption of zinc in the intestines; (3) Zinc binding to blood albumin and red blood cells is another important aspect to consider; (4) Cytosolic zinc buffering and muffling of intra- and extracellular zinc, which can lead to a marked reduction in free zinc plasma concentration [184] (Figure 3).

As shown in Figure 3, oral zinc bioavailability is affected by dietary chelating effects of phytates, iron, copper, fibre, and other factors in the small intestine, making oral zinc an unpredictable delivery method [185]. Specific transporters found in the apical membrane of enterocytes absorb zinc in the duodenum and the first part of the jejunum [186]. When it is absorbed, zinc is transported from the lumen into the enterocytes and released in a rate-limiting step into the portal blood, where it is predominantly bound to albumin, which distributes the metal in the body. Metallothionein controls zinc intestinal absorption whereas zinc regulates metallothionein gene expression [187]. Under conditions of oxidative stress, zinc is liberated from its complex with metallothionein and redistributed within cells to carry out antioxidant functions [188]. The zinc-dependent transcription factor MTF-1 is responsible for activating the expression of metallothionein genes, providing cellular protection against oxidative stress. MTF-1′s regulation is linked to the prevailing free zinc concentration in the cellular environment, where it acts as an intracellular zinc sensor, maintaining cytosolic free zinc ion levels at low picomolar to nanomolar concentrations [189]. However, due to the release of zinc from proteins and organelles, zinc concentrations can transiently and locally increase. This enables zinc to impact gene expression, enzymatic activity, and cell signalling [190]. It also plays a role in balancing the immune response, functioning as a “gatekeeper” [191]. Mitochondria, endoplasmic reticulum, and the Golgi apparatus sequester excess zinc to maintain a low steady-state level of free Zn^2+^ [192].

When in blood, albumin, which presents a high zinc buffering capacity and is responsible for up to 98% of the exchangeable component of Zn^2+^ found in plasma, effectively acts as an extracellular “zinc buffer” acting as a defence mechanism against a sudden increase in blood zinc levels [193]. Serum albumin likely regulates the uptake of Zn^2+^ into cells, such as endothelial cells and erythrocytes, potentially through mechanisms like endocytosis [194]. This function acts as a protective measure for blood cells and the endothelial cells that line blood vessels, shielding them from the potentially harmful levels of Zn^2+^ found in the plasma [195]. The balance of zinc within cells is meticulously managed through the expression of zinc “importers” known as ZIP 1-14 [191]. These importers raise cytoplasmic zinc levels by facilitating the import of zinc [196]. Conversely, ZnT 1-10, functioning as zinc “exporters,” lower cytoplasmic zinc levels by facilitating zinc export [197]. When B and T cells are stimulated, there is a sustained increase in intracellular zinc. This increase is primarily a result of the downregulation of ZnT1, ZnT 4-7, and the upregulation of ZIP6, ZIP8, and ZIP10 [198]. Conversely, the upregulation of ZnT1, located in cellular membranes, enables the removal of excess zinc from the cell. [199].

In addition, red blood cells sequester more than 90% of the extracellular zinc with an efflux rate of less than 2% compared to the influx rate, counteracting unpredictable increases of Zn^2+^ ions inside the tissue cells [183]. Zinc transporters and sensors localised to cytoplasmic and organellar membranes buffer and muffle the cytosolic zinc ion by either transiently shuttling Zn^2+^ ions into subcellular stores or by removing Zn^2+^ ions from the cell [200] (Figure 3).

## 6. Plasma Concentrations in Critical Illness, Sepsis, and COVID-19

Numerous experimental [201,202,203] and clinical studies have shown an association between low serum zinc concentration and critical illness [204,205,206]. During critical illness, plasma zinc value concentrations decrease rapidly due to a combination of deregulated intakes [206], excretion [207], and zinc redistribution into the cellular compartment [208]. Zinc concentrations are mostly substantially lower in patients with sepsis [209], COVID-19 [210], burn injury, and/or acute respiratory distress syndrome (ARDS) [211] compared to healthy people [212,213]. Plasma zinc concentration decreases with increased IL-6, IL-8, TNF-α levels and higher SOFA scores [214], decreases whenever CRP exceeds 20 mg/L, and normalizes with the resolution of inflammation [207]. This connection arises because of the shift of zinc from plasma albumin to the liver, where it binds to the elevated levels of metallothionein. This situation adds complexity to the interpretation of plasma zinc levels [215]. The liver’s metallothionein levels increase in response to cytokine release linked to inflammation, leading to the sequestration of zinc within the liver [216]. Additionally, the inflammatory response induces higher zinc excretion, which contributes to a further reduction in plasma zinc levels [206]. Accordingly, simultaneous acute phase inflammatory response biomarkers measurements should be assessed along with plasma zinc levels [10]. However, while analysis and interpretation of plasma zinc concentrations require consideration of several other factors, such as inflammation [217,218], 12% of ICU clinicians surveyed reported that they measure zinc plasma concentrations once per week [219]. Importantly, reduced activity of fatty acid desaturases 1 and 2 (FADS1/2), have been proposed as candidate biomarkers for assessing zinc status and the effectiveness of low-dose zinc interventions in these patients [220].

Decreased serum levels of zinc are associated with increasing severity of cardiovascular organ dysfunction [221], higher dosages of vasopressors [211], and apoptosis of airway epithelial cells [222] in critically ill patients who developed ARDS [223]. In observational studies, reductions in plasma zinc concentrations have been associated with the development of ARDS (4.4 μM in ARDS and 5.9 μM in sepsis compared to 8.3 μM in ICU controls, *p* < 0.001) [223], sepsis (recurrent 4.8 μM vs. no recurrent sepsis episode 7.16 μM in ICU controls day 3, *p* < 0.05) [224], and increased mortality (40.9% were below 638.7 μg/L, the threshold for Zn deficiency compared to discharged and controls, *p* < 0.001) [225].

Recent studies have shown that 31% [226] to 57.4% [227] of COVID-19 patients have decreased zinc plasma concentrations compared to healthy controls (74.5 (53.4–94.6) μg/dL vs. 105.8 (95.65–120.90) μg/dL (*p* < 0.001) [227]. It has been suggested that the low zinc levels in COVID-19 patients may represent an acute phase reaction due to COVID-19 infection [228,229]. The cytokine storm redistributes zinc from the plasma into cells as a protective measure to prevent immune dysfunction, systemic inflammation, oxidative stress that can lead to DNA damage, and cell apoptosis [210]. Additionally, the observed low zinc levels may result from a combination of a pre-existing deficiency, altered protein binding, increased losses, heightened metabolic demand, and dilution secondary to fluid overload [184]. These low zinc levels also contribute to immune dysfunction, which, in turn, increases angiotensin-converting enzyme 2 (ACE2) activity, potentially facilitating binding with SARS-CoV-2. In vitro, it has been observed that the spike protein of the SARS-CoV-2 virus interacts with ACE2 and the serine protease transmembrane protease serine 2 (TMPRSS2) in the alveoli, allowing the virus to enter host cells. ACE2 is an enzyme that relies on zinc for its function and consists of two subdomains, I and II, specifically involved in zinc binding. Prior research has indicated that a decrease in zinc levels promotes the interaction between ACE2 and the spike protein of SARS-CoV-2 (Figure 4). The RNA synthesis of SARS-CoV-2 is catalysed by an RNA-dependent RNA polymerase (RdRp), a core enzyme within its multiprotein replication and transcription complex [184]. During the acute phase of COVID-19, the inhibitory effect of zinc on virus entry, fusion, RNA synthesis, replication, protein translation, and viral dissemination is compromised due to insufficiently low zinc levels [230].

These patients develop more complications, need ICU admission (standard error 0.566, 95% CI 0.086 to 0.790, *p* = 0.017) [231], endotracheal intubation, and mechanical ventilation [231] more often, have a prolonged hospital stay, and increased mortality [227]. In critically ill COVID-19 patients, the persistent presence of low serum zinc levels showed an inverse correlation with disease severity, inflammatory markers [210], the occurrence of a “cytokine storm” [86], the development of long COVID, and mortality [224]. The symptoms of long COVID have their origins in neuroimmune and neuro-oxidative factors and are partially driven by an imbalance between oxidants and antioxidants, reduced glutathione peroxidase levels, and lower zinc concentrations [219]. Additionally, there is an increase in myeloperoxidase (MPO) and NO production, along with the formation of aldehydes associated with lipid peroxidation [220]. Furthermore, given that zinc plays a role in reducing pro-inflammatory Th17 cells, a deficiency in zinc might lead to endothelial dysfunction, increased susceptibility to autoimmune conditions, and the emergence of long COVID symptoms [4]. Consequently, zinc supplementation holds the potential to alleviate long COVID symptoms, particularly issues like fatigue and reduced sleep quality that are linked to inflammation [221].

## 7. Clinical Studies

### 7.1. Clinical Studies in Sepsis

Currently, there are no completed RCTs that investigated zinc as monotherapy in sepsis. There is only one ongoing trial of zinc supplementation in mechanically ventilated patients with severe sepsis (ClinicalTrials.gov Identifier: NCT01162109, Estimated Study Completion Date: December 2023), which limits the capacity to describe the effect of zinc supplementation. This is a randomized phase-I dose-finding study, examining the pharmacological dosing of IV zinc, its pharmacokinetics, impact of zinc on inflammation, immunity, oxidant defence, and toxicity. Although the use of pharmacologic zinc supplementation for low zinc levels is not supported by the existing evidence [10], 3% of ICU clinicians routinely prescribe zinc supplementation [219]. Furthermore, most RCTs that have evaluated zinc supplementation in sepsis have studied zinc as one of the more frequently administered micronutrients [232]. A combination of “selenium, zinc and copper” was ranked the best for lowest ICU stay; a combination of “selenium, zinc, copper and vitamin E” was ranked the best for infection risk reduction; a combination of “selenium, zinc and manganese” best for reduced ventilator days; “selenium, zinc and copper” was ranked the best for reduced hospital stay [232]. This problem has been highlighted previously by recognizing that nutrition is more than the sum of its parts and arguing that a micronutrient deficiency in sepsis might not just be a “body humours” imbalance but a sepsis “epiphenomenon” [233,234].

Regarding the paediatric population, in a double-blind, placebo-controlled trial, 352 infants aged 7–120 days with probable serious bacterial infection were randomly assigned to receive either 10 mg of zinc or placebo orally every day in addition to standard antibiotic treatment at three hospitals in New Delhi, India. Fewer treatment failures occurred in the zinc group (treatment of 15 infants with zinc would prevent one treatment failure) [235]. A phase 2 RCT in 72 children with severe sepsis who received either oral zinc sulfate at a dose of 10 to 20 mg daily for 10 days or no zinc has not reported any results yet [236]. In Pediatric Critical Illness Stress-Induced Immune Suppression (CRISIS), zinc, selenium, glutamine, and intravenous metoclopramide, compared with whey protein supplementation, conferred no advantage in the immune-competent population [237].

### 7.2. Clinical Studies in COVID-19

#### 7.2.1. Pharmacokinetic Results and Dose-Finding

A recent study found that correcting zinc deficiency in hospitalized COVID-19 patients, defined as a serum zinc level less than 70 μg/dL (equivalent to 10.7 µmol/L), with enteral delivery of 100–150 mg/day elemental zinc in the form of zinc acetate, takes between 14 and 24 days [226]. Prasad and colleagues have reported that zinc supplementation might result in a milder form of COVID-19, as zinc inhibits the pH-dependent stages of SARS-CoV-2 replication within cells, leading to an increase in the pH of intracellular vesicles [238]. The need to use significantly higher zinc doses than the recommended upper daily oral intake is rooted in Hambidge’s analysis, which indicates that the maximal absorption (Amax) for zinc through oral delivery is limited to just 7 mg Zn/day [182]. This limited Amax is attributed to the complex molecular processes involved in the absorption of oral zinc into and across the enterocyte and basolateral membrane. Given the low bioavailability of orally administered zinc, high-dose intravenous zinc has been proposed as the preferred method of administration [239].

#### 7.2.2. Completed RCTs with Clinical Outcome

COVID-19 RCTs registered in ClinicalTrials.gov differ regarding dosing, route, or chemical form, have a non-random or non-blinded design, studied heterogeneous populations with individual variability [240], report selected endpoints, add other drugs (hydroxychloroquine, azithromycin, vitamin C, D, resveratrol ivermectin, doxycycline, tocilizumab, dexamethasone, famotidine nitazoxanide, ribavirin, darunavir, ritonavir, lactoferrin, nitazoxanide, quercetin, or bromelain, oral methylene blue solution, inhaled phenformin, and selenium micronutrients) [241,242], have not yet been completed. Using zinc supplementation alone in treating COVID-19, two RCTs have reported results on clinical outcomes. One RCT reported that zinc supplementation (oral zinc 50 mg per day) improved mortality at 30 days (6.5% in the zinc group and 9.2% in the placebo group (OR: 0.68; 95% CI 0.34–1.35)) and decreased ICU admission rates (5.2% vs. 11.3% (OR: 0.43; 95% CI 0.21–0.87)) and LOS (difference: 3.5 days; 95% CI 2.76–4.23 in the inpatient group) [243]. In a separate clinical trial conducted as a randomly assigned 2 × 2 factorial placebo-controlled study with a 1:1:1:1 allocation ratio, which included nonpregnant adults with COVID-19 from hospitals in Mumbai and Pune, India, no significant impact of zinc supplementation alone was observed. This included its effect on the duration of specific symptoms, the requirement for mechanical ventilation, length of hospital stay, all-cause mortality, as well as various blood biomarkers encompassing nutritional, inflammatory, and immunological markers [244]. A completed RCT in Tunisia has not posted results yet. This study hypothesized that zinc supplementation (25 mg of zinc twice a day for 15 days) would reduce 30-day mortality and the need for ICU admission among non-critically ill patients with COVID-19 [245]. In another completed RCT of a parallel assignment model in Spain, zinc supplementation (75 mg Zn element) was administered in the acute phase of SARS-CoV-2 infection for 14 days in the intervention vs. no intervention group. No results have been reported on the monitored responses on viral loads, inflammatory and novel zinc-related clinical markers, and SARS-CoV-2-specific T- and B-cell populations [246]. In a Phase IIa double-blinded, randomized, controlled trial in hospitalized COVID-19 patients, high-dose intravenous zinc, but not placebo, increased serum zinc levels above the deficiency cut-off of 10.7 μmol/L within one week. No adverse events were observed, but the study did not reach its target enrolment to assess clinical outcome endpoints [228]. An RCT from St. Francis Hospital, New York aiming to assess the efficacy of zinc (zinc Sulfate 220 mg once daily for 5 days) in a higher-risk COVID-19-positive outpatient population has ended up having enrolled 3 participants [247].

#### 7.2.3. Currently Ongoing Trials

A placebo-controlled, randomized, parallel-group, adaptive, phase 2 study has been set in June 2023 to evaluate the safety and efficacy of liposomal zinc, liposomal bovine lactoferrin, and standard of care compared to placebo and standard of care in non-hospitalized patients with COVID-19 for 10 days [248]. The primary endpoint is the reduction in the signs and symptoms of COVID-19 from baseline to 28 and 60 days after the last dose.

As of 18 September 2023, 25 RCTs were found after searching for Zn and COVID-19 at the clinical trial website https://clinicaltrials.gov/ct2/home, accessed on 18 September 2023. Completed or ongoing RCTs with only zinc supplementation are reported in Table 1.

## 8. Conclusions

Zinc has an immunomodulatory role in sepsis, a life-threatening organ dysfunction caused by a dysregulated host response to infection, with a high mortality rate. In addition, zinc is a potent inhibitor of caspases-3, -7, and -8, modulating the caspase-controlled apoptosis, necroptosis, and inflammatory processes in COVID-19 and sepsis. Lack of zinc negatively affects various immune system functions, such as phagocytosis, oxidative burst, degranulation, cytokine production, and chemotaxis. While plasma zinc levels decline rapidly during COVID-19 and sepsis, this reduction may not necessarily indicate a genuine zinc deficiency. Instead, it could be an outcome of an inflammatory response, characterized by an increase in hepatic metallothionein triggered by the release of cytokines associated with inflammation and oxidative stress. Also, bioavailability mechanisms affect the enterocytes’ uptake of zinc and the binding to the blood albumin or red blood cells, and cytosolic zinc buffering and muffling. Randomized controlled clinical trials are still lacking in this area, while the existing evidence does not support pharmacological zinc supplementation in patients with COVID-19 or sepsis. Understanding the pharmacological mechanism of zinc and the programmed cell death caused by invasive infections may indicate potential clinical applications of zinc in COVID-19 and sepsis.

## 9. Future Perspective

With the growing significance of understanding the molecular mechanisms of zinc, it is imperative to thoroughly investigate its potential for future therapeutic approaches in the context of pandemics and sepsis [249]. To further elucidate the role of zinc in different COVID-19 subtypes and sepsis, forthcoming studies should concentrate on immune cells and endothelial cells. Such research aims to ascertain whether these cells can be directly targeted for COVID-19 treatment, potentially delaying the onset of sepsis and expanding the treatment options available. Since the immune response is markedly associated with the early pathological progression of both COVID-19 and sepsis [250], whether zinc is readily available to support the normal functioning of immune cells following these life-threatening disorders may be the first thing to consider in the near term.

To address the challenge of excessive lysosomal accumulation of hidden bacteria or viruses, experimental nanoparticles referred to as “Alpha-MOFs” have been developed. These nanoparticles specifically target malfunctioning monocytes and macrophages to actively enhance phagocytosis by releasing lysosome-sensitive ions from a mineralized metal-organic framework (MOF) [88,89]. Alpha-MOFs have demonstrated their ability to restore immune function and improve survival in sepsis. They exhibit excellent stability and biosafety in peripheral blood and efficiently release calcium and zinc ions into monocyte/macrophage lysosomes, facilitating the phagocytosis and degradation of bacteria [87].

Given that intravenous zinc administration can bypass the various factors affecting its bioavailability and effectively reach its target, there is a need for randomized clinical trials to investigate the potential benefits of zinc supplementation as an adjunctive treatment in sepsis and COVID-19 [251]. Given that intravenous zinc administration can bypass the various factors affecting its bioavailability and effectively reach its target, there is a need for randomized clinical trials to investigate the potential benefits of zinc supplementation as an adjunctive treatment in sepsis and COVID-19 [252].

## Figures and Tables

**Figure 1 antioxidants-12-01942-f001:**
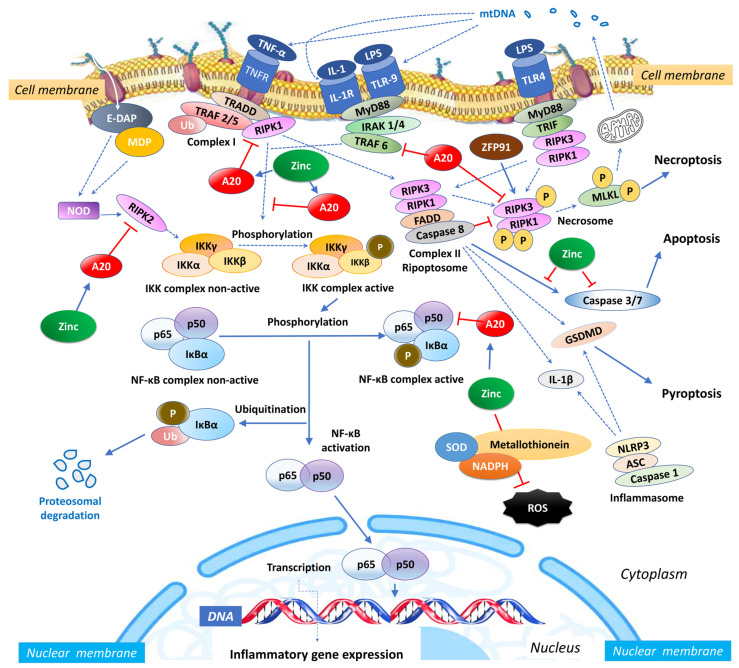
Intracellular zinc and A20 regulatory activities in sepsis. NF-κB: nuclear factor κB; TNF: tumour necrosis factor; TNFR: tumour necrosis factor receptor; TRAF: TNF receptor-associated factor; TRADD: death domain-containing adaptor protein; E-DAP: meso-diaminopimelic acid; MDP: muramyl dipeptide; RIPK: receptor-interacting serine/threonine-protein kinase; IKK: inhibitor of κB kinase; IκBα: nuclear factor of kappa light polypeptide gene enhancer in B-cells inhibitor, alpha; Ub: ubiquitin; SOD: superoxide dismutase; NADPH: nicotinamide adenine dinucleotide phosphate; NLRP3: NLR Family Pyrin Domain Containing 3; ASC: apoptosis-associated speck-like protein containing a caspase recruitment domain; IL: interleukin; MLKL: mixed lineage kinase domain like pseudokinase; MyD88: myeloid differentiation primary response 88; IRAK-4: interleukin-1 receptor-associated kinase 4; ZFP91: zinc finger protein 91; LPS: Lipopolysaccharide; mtDNA: mitochondrial DNA; FADD: Fas-associated death domain; TLR: toll-like receptor; TRIF: TIR-domain-containing adaptor-inducing IFN-β; ROS: reactive oxygen species.

**Figure 2 antioxidants-12-01942-f002:**
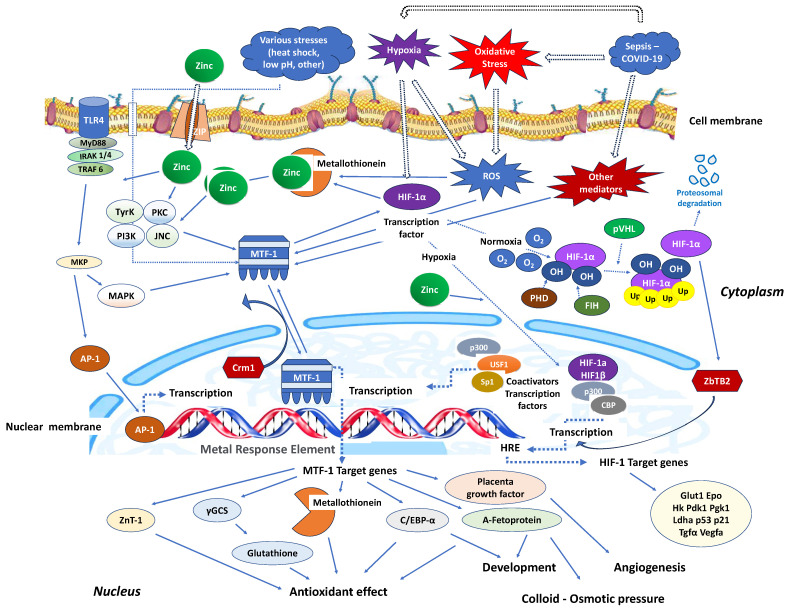
Critical zinc effects on transcriptional factors apart from NF-κB. MTF-1: Intracellular metal responsive transcription factor 1; HIF-1α: hypoxia-inducible factor 1-alpha; AP-1: activator protein-1; MRE: metal-response element; PHD: prolyl hydroxylase domain enzyme; FIH: factor-inhibiting HIF; Crm1: chromosomal maintenance 1; O_2_: oxygen molecule; HRE: hypoxia response element; p300: protein 300; CBP: CREB-binding protein; ZBTB2: zinc finger and bric-a-brac (BTB)-domain-containing protein 2; OH: hydroxyl group; Ub: ubiquitin; ZIP: Zinc importer protein; pVHL: von Hippel–Lindau protein, ROS: reactive oxygen species; MyD88: myeloid differentiation primary response 88; IRAK-4: interleukin-1 receptor-associated kinase 4; TLR: toll-like receptor; TRAF: TNF receptor-associated factor; MKP: MAP kinase phosphatases; MAKP: mitogen-activated protein kinase; USF1: upstream transcription factor 1; γGCS: γ-glutamylcysteine synthetase; Glut1: Glucose transporter 1; Hk: hexokinase; PdK1: pyruvate dehydrogenase kinase-1; PgK1: phosphoglycerate kinase-1; Ldha: lactate dehydrogenase A; Tgfa: transforming growth factor, alpha; Vegfa: vascular endothelial growth factor A; C/EBV-α: CCAAT/enhancer binding protein alpha.

**Figure 3 antioxidants-12-01942-f003:**
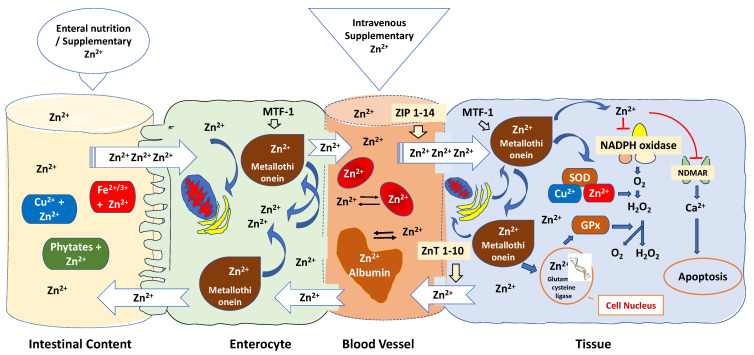
Regulatory mechanisms that affect zinc bioavailability are related to antioxidant mechanisms. Oral zinc bioavailability is affected by dietary chelating effects of phytates, iron, copper, fibre, and other factors in the small intestine, making oral zinc an unpredictable delivery method. Specific transporters found in the apical membrane of enterocytes absorb zinc, released in a rate-limiting step into the portal blood. Metallothionein regulates the absorption of zinc in the intestines, while zinc, in turn, modulates the gene expression of metallothionein. The zinc-dependent transcription factor MTF-1 functions as an internal zinc sensor, ensuring that cytosolic free zinc ion concentrations are maintained at low picomolar to nanomolar levels. When zinc is released from proteins and organelles, it can temporarily and locally raise zinc concentrations, influencing gene expression, enzymatic activity, and cell signalling, whereas mitochondria, endoplasmic reticulum, and the Golgi apparatus sequester excess zinc to maintain a low steady-state level of free Zn^2+^. Zinc homeostasis is controlled through the expression of zinc “importers” (ZIP 1-14), which increase cytoplasmic zinc levels by importing zinc and zinc “exporters” (ZnT 1-10), which reduce cytoplasmic zinc levels via zinc export. Zinc transporters and sensors localised to cytoplasmic and organellar membranes buffer and muffle the cytosolic zinc ion by either transiently shuttling Zn^2+^ ions into subcellular stores or by removing Zn^2+^ ions from the cell. Albumin serves as an extracellular “zinc buffer,” effectively acting as a protective shield against abrupt surges in blood zinc levels. Furthermore, red blood cells capture over 90% of extracellular zinc, countering unanticipated rises in Zn^2+^ ions within tissue cells. Zn: Zinc; Cu: Copper; Fe: Iron; MTF-1: Metal-responsive transcription factor 1; ZIP: Zinc importer protein; ZnT: Zinc transporter; SOD: superoxide dismutase enzyme; GPx: Glutathione peroxidase; NADPH: nicotinamide adenine dinucleotide phosphate; NDMAR: N-methyl-D-aspartate receptor.

**Figure 4 antioxidants-12-01942-f004:**
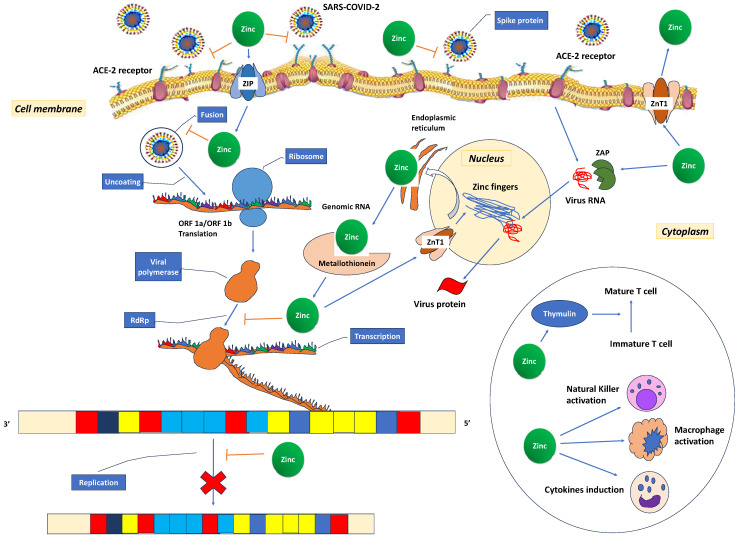
Possible mechanisms for zinc’s role in the treatment of COVID-19. The spike proteins of SARS-CoV-2 attach to ACE2 receptors, facilitating the release of the RNA genome into the host cell and subsequent translation of both structural and non-structural proteins. Open reading frames (ORF)1a and ORF1ab are translated, giving rise to pp1a and pp1ab polyproteins, which are subsequently cleaved by proteases encoded by ORF1a, leading to the formation of non-structural proteins. This process is succeeded by the assembly and budding of virions into the endoplasmic reticulum-Golgi intermediate compartment (ERGIC), ultimately leading to the release of virions from the infected cell through exocytosis. Zinc could potentially exert its therapeutic effects in several ways. It may reduce ACE2 activity, a known receptor for SARS-CoV-2, and hinder the interaction between SARS-CoV-2 and this receptor, consequently inhibiting the fusion of SARS-CoV-2 with the host cell membrane. Additionally, zinc might exhibit antiviral properties by inhibiting the RNA-dependent RNA polymerase (RdRp) and reducing its ability to bind to the viral template. Zinc, through zinc finger proteins, may deactivate virus RNA in the nucleus and the endoplasmic reticulum. In the circle, the activating effect of zinc in various immune cells is depicted. SARS-CoV-2: severe acute respiratory syndrome coronavirus 2; ACE2: angiotensin-converting enzyme 2; ORF: open reading frames; RdRp: RNA-dependent RNA polymerase. ZAP: zinc finger antiviral protein; ZnT: Zinc transporter; ZIP: Zinc importer protein.

**Table 1 antioxidants-12-01942-t001:** Randomized Controlled Trials on zinc supplementation in COVID-19.

Type of Study	Patient Population	Intervention	Clinical Endpoints	Surrogate Endpoints
Randomized Double-Blind Controlled Trial ClinicalTrials.gov, accessed on 18 September 2023, NCT05212480 [243]	470 not admitted to ICU and having had a symptom onset ≤ 7 days. COVID-19 patients 190 (40.4%) ambulatory and 280 patients (59.6%) hospitalized.	Oral zinc (50 mg per day, given in two doses) (*n* = 231) or matching placebo (*n* = 239) for 15 days.	Significantly lower mortality at 30 days (6.5% vs. 9.2%), ICU admission rates (5.2% vs. 11.3%). LOS (difference: 3.5 days), duration of COVID-19 (difference: 1.9 days).	No difference in nutritional, inflammatory, and immunological bio markers. No adverse effects.
Double-Blind Randomized Placebo-Controlled Trial 2 × 2 factorial placebo-controlled trial with 1:1:1:1 allocation ratio ClinicalTrials.gov accessed on 18 September 2023, NCT04641195 [244]	Nonpregnant adults with COVID-19 Participants (*n* = 181).	Randomly assigned to zinc (40 mg daily), vitamin D3, vitamin D3 and zinc, or placebo, for 8 wk.	No effect of zinc alone on duration of individual symptoms, need for mechanical ventilation, LOS, all-cause mortality.	No effect on nutritional, inflammatory, immunological biomarkers and no adverse events. Nonsignificant increases in serum zinc at endline following zinc supplementation.
Phase IIa double-blind, randomized controlled trial Australia New Zealand Clinical Trial Register Registration no. ACTRN12620000454976 [228]	Hospitalized patients (33) with COVID-19 with SpO_2_ of 94% or less while on ambient air. Randomized to either HDIVZn (*n* = 15) or placebo (*n* = 18).	Zinc Chloride (ZnCl_2_) diluted in 250 mL of normal saline and infused via peripheral intravenous access over 3 h at a dose of 0.5 mg/kg/day (elemental zinc concentration, 0.24 mg/kg/day) for a maximum of 7 days.	The study did not reach its target enrolment to assess the primary outcome (level of oxygen flow, lowest PaO_2_/FiO_2_ ratio in ventilated patients.	HDIVZn, but not placebo, increased serum zinc levels above the deficiency cut off of 10.7 µmol/L (*p* < 0.001) on Day 6. No serious adverse events.
ClinicalTrials.gov, accessed on 18 September 2023. Phase 4 ID NCT04621461 (Completion Feb 8, 2021) [247]	Ambulatory only non-hospitalized—3 participants.	Dietary Supplement: Zinc Sulphate 220 MGDrug: Placebo.	No results posted on COVID-19 related complications that require an emergency room visit, ICU admission, mortality.	No results—study has ended.
RCT of parallel assignment model—phase 4 ClinicalTrials.gov Study completion, accessed on 18 September 2023. 2022-05-25 ID NCT05778383 [246]	SARS-CoV-2 infection requiring hospital admission. Actual enrolment 75 patients randomized 1:1 to either zinc or no intervention (Standard of Care).	Each participant allocated in the intervention arm will be treated as Standard of Care and will be supplemented with 240 mg of Zinc Acetate (75 mg Zn element) vs. standard of care group.	No results posted on disease progression and clinical outcomes—study has completed.	No results reported on viral loads, inflammatory and novel zinc-related clinical markers, and SARS-CoV-2-specific T- and B-cell.
RCT of parallel assignment model ClinicalTrials.gov, accessed on 18 September 2023. Study completion 2022-05-04 ID NCT04621461 [245].	Non-critically ill patients with COVID-19 in Tunisia. Actual enrolment 460 patients randomized 1:1 to either zinc or placebo.	Dietary Supplement: Zinc at a dose of 25 mg twice a day for 15 days vs. Placebo.	No results posted yet on 30-day mortality and need to ICU admission among non-critically ill patients with COVID-19.	No results—study has completed.
On-going randomized, double-blind, placebo-controlled trial Phase 2 Estimated study completion 2024-11 NCT05783180 [248]	Estimated enrolment 40 patients with mild to moderate SARS-CoV-2 infection. Subjects will be randomized in a 1:1 ratio to either treatment, *n* = 20, or control group, placebo, *n* = 20.	Treatment group: Liposomal zinc (20 mg/20 mL QD), liposomal bovine lactoferrin (64 mg/20 mL, TID), and standard of care compared to placebo and standard of care in non-hospitalized patients with COVID-19 for 10 days.	Primary endpoint reduction in the signs and symptoms of COVID-19 from baseline to 28 and 60 days after the last dose.	Not yet recruiting.

ICU: Intensive Care Unit; LOS: length of stay; HDIVZn: high dose intravenous zinc.

## Data Availability

Not applicable.

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
