# Peer review of "The Anti-Oxidative, Anti-Inflammatory, Anti-Apoptotic, and Anti-Necroptotic Role of Zinc in COVID-19 and Sepsis"

_antioxidants, 2023, doi:10.3390/antiox12111942_

Round 1

Reviewer 1 Report

Comments and Suggestions for Authors

ID: antioxidants-2654498

The anti-oxidative anti-inflammatory anti-apoptotic and anti-necroptotic role of Zinc in COVID-19 and sepsis. By Briassoulis et al.

To the Authors:

General comments:

The authors reviewed the immunomodulatory role of zinc in sepsis and COVID-19.  They found that zinc is a potent inhibitor of caspases -3, -7, and -8, modulating the caspase-controlled apoptosis, necroptosis, and inflammatory processes, while randomized controlled clinical trials regarding the use of pharmacological zinc supplementation in patients with COVID-19 or sepsis are relatively scarce.  It was considered that this review is described well and summarizes important points; however, several points should be addressed to improve the manuscript.

Specific comments:

1. Please provide a discussion about the molecular mechanisms by which plasma zinc levels were decreased during the acute phase of COVID-19 (lines 601-610).

2. The interrelationship between serum zinc levels and the chronic phase of COVID-19, namely long COVID, could also be discussed.

3. COVID-19 affects not only inflammation and the immune system but also other pathophysiological changes, including hormones.  Please add the description regarding the relationship between zinc and hormonal changes in COVID-19.

4. For the wide-range of readers, it would be nice to produce a schema to show the interrelationship between the COVID-19 and zinc condition.

Author Response

Reviewer comments

First, we are very grateful to the reviewer for the suggestions that we believe have contributed to our manuscript's remarkable improvement. The following is a list of point-to-point responses to the proposed comments. Changes are shown in blue here and in the revised MS. The lines correspond to simple mark-up track changes. In case of full mark-up track changes, numbering is changed.

Reviewer #1:

To the Authors:

General comments:

The authors reviewed the immunomodulatory role of zinc in sepsis and COVID-19. They found

that zinc is a potent inhibitor of caspases -3, -7, and -8, modulating the caspase-controlled apoptosis, necroptosis, and inflammatory processes, while randomized controlled clinical trials

regarding the use of pharmacological zinc supplementation in patients with COVID-19 or sepsis

are relatively scarce. It was considered that this review is described well and summarizes

important points; however, several points should be addressed to improve the manuscript.

  • Response

Thank you for your kind words.

Specific comments:

  1. Please provide a discussion about the molecular mechanisms by which plasma zinc levels

were decreased during the acute phase of COVID-19 (lines 601-610).

  • Response

Added. Lines 742-7592. The cytokine storm redistributes zinc from the plasma into cells as a protective measure to prevent immune dysfunction, systemic inflammation, oxidative stress that can lead to DNA damage, and cell apoptosis [210]. Additionally, the observed low zinc levels may result from a combination of a pre-existing deficiency, altered protein binding, increased losses, heightened metabolic demand, and dilution secondary to fluid overload [184]. These low zinc levels also contribute to immune dysfunction, which, in turn, increases ACE-2 activity, potentially facilitating binding with SARS-CoV-2. In vitro, it has been observed that the spike protein of the SARS-CoV-2 virus interacts with angiotensin-converting enzyme 2 (ACE2) and the serine protease transmembrane protease serine 2 (TMPRSS2) in the alveoli, allowing the virus to enter host cells. ACE2 is an enzyme that relies on zinc for its function and consists of two subdomains, I and II, specifically involved in zinc binding. Prior research has indicated that a decrease in zinc levels promotes the interaction between ACE2 and the spike protein of SARS-CoV-2 (Figure 4). The RNA synthesis of SARS-CoV-2 is catalysed by an RNA-dependent RNA polymerase (RdRp), a core enzyme within its multiprotein replication and transcription complex [184]. During the acute phase of COVID-19, the inhibitory effect of zinc on virus entry, fusion, RNA synthesis, replication, protein translation, and viral dissemination is compromised due to insufficiently low zinc levels [231].

  1. The interrelationship between serum zinc levels and the chronic phase of COVID-19, namely

long COVID, could also be discussed.

  • Response

Added. Lines 782-795. In critically ill COVID‐19 patients, the persistent presence of low serum zinc levels showed an inverse correlation with disease severity, inflammatory markers [210], the occurrence of a "cytokine storm" [86], the development of long COVID, and mortality [233]. The symptoms of long COVID have their origins in neuroimmune and neuro-oxidative factors and are partially driven by an imbalance between oxidants and antioxidants, reduced glutathione peroxidase levels, and lower zinc concentrations [218]. Additionally, there is an increase in myeloperoxidase (MPO) and NO production, along with the formation of aldehydes associated with lipid peroxidation [219]. Furthermore, given that zinc plays a role in reducing pro-inflammatory Th17 cells, a deficiency in zinc might lead to endothelial dysfunction, increased susceptibility to autoimmune conditions, and the emergence of long COVID symptoms [4]. Consequently, zinc supplementation holds the potential to alleviate long COVID symptoms, particularly issues like fatigue and reduced sleep quality that are linked to inflammation [220].

  1. COVID-19 affects not only inflammation and the immune system but also other

pathophysiological changes, including hormones. Please add the description regarding the

relationship between zinc and hormonal changes in COVID-19.

  • Response

Added. Lines 415-444. The pituitary gland boasts the most substantial zinc concentration, which augments the performance of pituitary hormones [103]. When there is a shortage of zinc, it leads to a deficiency in the secretion of pituitary growth hormone (GH), nerve growth factor (NGF), insulin-like growth factor-1 (IGF-1) [104], and hippocampal nerve growth factor-7S (NGF-7S) [105]. Importantly, COVID symptoms might also be due to hypothalamic-pituitary-adrenal (HPA) axis suppression induced by the virus and the immune system. The angiotensin receptor converting enzyme 2 (ACE2) expression in the zona fasciculata and zona reticularis of the adrenal cortex, injured by the identified in autopsies COVID-19 virus, could affect glucocorticoid synthesis [106]. ACE2 is expressed not only in the lower respiratory tract but also in the somatosensory cortex, rectal/orbital gyrus, temporal lobe, hypothalamus/thalamus, brainstem, and cerebellum. Direct viral infection of the neuronal cells in these regions occurs by the virus binding to the ACE2 receptor, resulting in demyelination and neurodegeneration and inducing a greater risk of long‐term effects in some patients recovering from COVID‐19 [107]. Zinc deficiency and genetic susceptibility, also contribute to neurodegeneration [108]. Thus, the elevated zinc deficiency gene for the odorant metabolizing enzyme UDP-glucuronosyltransferase (UGT) has been linked to COVID-induced anosmia [109].

               While COVID-19 can induce stress-related metabolic disturbances and zinc deficiency can impact the proper functioning of insulin receptors on cell surfaces, the exact interactions between insulin resistance and zinc in the context of COVID-19 are not fully understood. In vivo, zinc improved the endothelium-dependent vasorelaxation, reversed the reduction of guanosine 5'-triphosphate cyclohydrolase 1 and tetrahydrobiopterin, and suppressed the elevation of ROS in streptozotocin-induced diabetic mice [110]. Finally, epigenetic changes on the male and female reproductive organs secondary to COVID-19 exposure are not well documented, although reports indicating adverse effects of COVID-19 on male and female gametogenesis are emerging [111]. SARS CoV-2 induces cytokine storm, increases ROS in the gametogenic cells, and depletes intracellular zinc, potentially resulting in oocyte and sperm damage. Furthermore, zinc deficiency mediating ROS overproduction potentiates oocyte and sperm damage, aggravating the effects of COVID-19 [111].

  1. For the wide-range of readers, it would be nice to produce a schema to show the

interrelationship between the COVID-19 and zinc condition.

  • Response

Produced. Line 762. Figure 4 was produced showing the interrelationship between COVID-19 and zinc condition.

Reviewer 2 Report

Comments and Suggestions for Authors

The paper describes the role of zinc in COVID-19 and sepsis. Zinc plays a crucial role in the human body, and its homeostasis is important for human health. The summary of the role of zinc in these diseases is novel; therefore, in my opinion, the manuscript is worth publishing in the Antioxidants journal.

Only minor suggestions:

Authors should carefully check the usage of abbreviations, e.g., the abbreviation 'RCD' is explained on page 7, while the abbreviation is first used in the Abstract and on page 2. A dot is missing in verse 52. 'In vitro' should be written using italics. On both figures, larger letters should be used.

Author Response

Reviewer comments

First, we are very grateful to the reviewer for the suggestions that we believe have contributed to our manuscript's remarkable improvement. The following is a list of point-to-point responses to the proposed comments. Changes are shown in blue here and in the revised MS.

Reviewer #2:

The paper describes the role of zinc in COVID-19 and sepsis. Zinc plays a crucial role in the

human body, and its homeostasis is important for human health. The summary of the role of zinc

in these diseases is novel; therefore, in my opinion, the manuscript is worth publishing in the

Antioxidants journal.

  • Response

Thank you for your kind words.

Only minor suggestions:

Authors should carefully check the usage of abbreviations, e.g., the abbreviation 'RCD' is

explained on page 7, while the abbreviation is first used in the Abstract and on page 2.

  • Response

Thank you, we now explain it on page 2. The abstract is usually thought to be a different part with its own abbreviations.

A dot is missing in verse 52. 'In vitro' should be written using italics.

  • Response

Thank you, we have now addressed and corrected all these (dot, italics, letters).

On both figures, larger letters should be used.

  • Response

Both figures have now been updated using larger letters and needed improvements.

Reviewer 3 Report

Comments and Suggestions for Authors

In their review manuscript, the authors focussed on the role of zinc in COVID-19 and sepsis patients. The authors provide some mechanistic insights to the functions of zinc, its antioxidant activity, and its role in inflammation, immune regulation, and cell death as well as its bioavailabiltity. This part is followed by plasma concentrations in critical illnes, sepsis, and COVID-19 patients. Finally, clincial studies in sepsis and COVID-19, which included aspects on zinc, are listed.

This narrative review is of general interest, However some concerns should be carefully addressed:

Line 51: Between “stress” and “Critically” a full stop is missing.

Lines 125/127: A gene does not encode GSH. Thus, Nrf2 cannot bind to the promoter region. Please correct.

Line 147: “beta” should be “B”.

Lines 157/158: Please specifiy how transcription factors can induce the production of ROS.

Lines 161/162: Which enzyme(s) phosphorylate(s) NF-kappaB?

Line 173: The abbreviation “ICAM-1” should be specified here (not in line 193).

Lines 174/175: Please specify how zinc affects binding of NF-kappaB to other TFs!

Line 189: How does zinc enhance I-kappaB stability? Please specify!

Line 206/210/220: Are the correct references [50], [51, 52], and [20] cited? Do these fit to the text?

Figure 1: In the Figure “Inflammaotory gene expression” should be changes to “Inflammatory gene expression”. How das mtDNA affect TNF? The A20 circle in the left upper part should be in front of the line. “Nucleus” might be better located below the DNA. The trancription factors have to bind to the DNA to induce transcription of target genes, don´t them? Possibly a further figure showing the effect of zinc on transcription factors apart from NF-kappaB would be nice!

Lines 245-248: How does zinc mediates these effects? This should be specified!

Part 4 “Regulated cell death” should be condensed to avoid a repetition in parts 4.1 and 4.2.

Line 363: Reference(s)?

Line 371: Between “[94]” and “Zinc” a full stop is missing.

Line 375: “(eryptosis)” should be “Eryptosis”.

Lines 427-429: Is there a word missing?

Line 437: Necrostatin-1, Necrostatin-1s, or both? Please specify!

Lines 476-479: Sentence doubled?

Line 571: “Numerous”? Reference [163] is only one study.

Line 576: “Acute Respiratory Distress Syndrome” should be “acute respiratory distress syndrome”

Part 6 “Plasma concentrations in critical illness, sepsis, COVID-19”: Is there a link to mechanisms, which are affected by low or high zinc in these patients?

Line 744: Zinc is not “expressed”.

Line 746: “hided” should be “hid” or “hidden”

Some parts of the manuscript do not have a link to zinc (lines 278-328, 484-493). These should be combined (see Part 4) or rewritten including aspects on zinc.

Author Response

Reviewer comments

First, we are very grateful to the reviewer for the suggestions that we believe have contributed to our manuscript's remarkable improvement. The following is a list of point-to-point responses to the proposed comments. Changes are shown in blue here and in the revised MS. The lines correspond to simple mark-up track changes. In case of full mark-up track changes, numbering is changed.

Reviewer #3: 

Suggestions for Authors
In their review manuscript, the authors focussed on the role of zinc in COVID-19 and sepsis
patients. The authors provide some mechanistic insights to the functions of zinc, its antioxidant
activity, and its role in inflammation, immune regulation, and cell death as well as its
bioavailabiltity. This part is followed by plasma concentrations in critical illnes, sepsis, and
COVID-19 patients. Finally, clincial studies in sepsis and COVID-19, which included aspects on
zinc, are listed. This narrative review is of general interest. However, some concerns should be carefully addressed. 

•    Response
Thank you for your kind words and inspired comments.

Line 51: Between “stress” and “Critically” a full stop is missing.

•    Response
Added. Line 51

Lines 125/127: A gene does not encode GSH. Thus, Nrf2 cannot bind to the promoter region.
Please correct.

•    Response
Corrected. Lines 126-130 …which regulates glutathione peroxidase 2 (GPX2), glutamate-cysteine ligase catalytic subunit (GCLC), thioredoxin reductase 1, members of the glutathione-S-transferase (GST), heme-oxygenase-1 (HO-1), and superoxide dismutase (SOD) by binding to the an-ti-oxidant responsive cis-acting enhancer sequence elements [20].

Line 147: “beta” should be “B”.

•    Response
Corrected. Line 149.

Lines 157/158: Please specify how transcription factors can induce the production of ROS.

•    Response
Specified. Lines 159-168. Various pro-inflammatory transcription factors, including NF-κB and activator protein-1 (AP-1), can induce the production of ROS, causing the release of inflammatory cytokines, which in turn enhance oxidative stress [33]. Cellular transcriptional response to ROS is mediated mainly by activation of mitogen-activated protein (MAP) kinases, cysteines Cys65 and Cys93 in redox effector factor (Ref-1), the redox-dependent modification of transcription factors p53, activating transcription factor/cAMP-response, element–binding protein (ATF/CREB), hypoxia-inducible factor (HIF)-1α, and HIF-like factor [34,35]. Superoxide anions undergo oxidative phosphorylation with the oxidative systems consisting of xanthine oxidase, nicotinamide adenine dinucleotide phosphate (NADPH) oxidases, 5-lipoxygenase, and cyclooxygenase [36]. 

Lines 161/162: Which enzyme(s) phosphorylate(s) NF-kappaB?

•    Response
Clarified. Lines 168-175. Strong experimental evidence shows that the proteasome partially processes phosphorylation of p105 and p100 precursor NF-κB proteins at S893 and S907 residues to produce NF-κB subunits p50 and p52. The p105 and p100 precursor proteins contain C-terminal ankyrin repeats conferring IκB-like function by sequestering interacting NF-κB subunits in the cytoplasm. The p50 NF-κB subunit is phosphorylated by protein kinase A (PKA) at S337, by checkpoint kinase 1 (Chk1) at 328 triggered by DNA damage, and by DNA-dependent protein kinase (DNA-PK) responded to TNF-α stimulus at S20 [37].

Line 173: The abbreviation “ICAM-1” should be specified here (not in line 193).

•    Response
Specified. Line 195.

Lines 174/175: Please specify how zinc affects binding of NF-kappaB to other TFs!

•    Response
Specified. Line 198-207. Upon zinc supplementation, zinc receptor metal regulatory transcription factor 1 (MTF-1) and other transcription factors responsible for cellular adaptation to oxidative stress enhance gene expression within the nucleus. This enhancement occurs by in-creasing DNA binding and recruiting various co-activators, thereby maintaining cellular homeostasis [43]. The release of zinc from metallothioneins activates NF-κB and influences its ability to transmit pro-inflammatory signals to the nucleus, regulating the expression of DNA for transcription factors involved in the inflammation process. Inflammatory states have shown a coordinated interplay among zinc, metallothioneins, MTF-1, and pro-inflammatory cytokines like IL-6 and TNF-α, orchestrated by the transcription factor NF-κB [44].

Line 189: How does zinc enhance I-kappaB stability? Please specify!

•    Response
Specified. Line 220-239. The activation of the NF-κB heterodimer, composed of p65-p50 and IκBα, occurs through the phosphorylation of IκB by the IκB kinase (IKK) complex. This leads to the ubiquitination and subsequent proteasomal degradation of IκB [47]. Zinc, through the inhibition of cyclic nucleotide phosphodiesterase (PDE), causes an increase in cyclic guanosine monophosphate (cGMP). This rise in cGMP activates protein kinase A (PKA) and, by phosphorylation, inhibits the protein kinase Raf-1. Moreover, zinc suppresses the activation of IKKβ induced by LPS [48]. Also, imported by the zinc transporter ZIP8, cytosolic zinc induces NF-κB inhibition downstream from mitogen-activated protein kinases (MAPKs) by blocking the IKK complex [49]. Finally, zinc negatively regulates the NF-κB pathway by inducing the expression of zinc-finger protein A20, which is the main negative regulator of the NF-κB activation [47]. Cytosolic zinc, transported via the zinc transporter ZIP8, induces the inhibition of NF-κB downstream of mitogen-activated protein kinases (MAPKs) by interfering with the IKK complex [46]. This, in turn, affects its target genes, such as TNF-α and IL-1-beta (IL-1β)[50], as well as chemoattractant proteins like monocyte chemoattractant protein-1 (MCP-1) and intercellular adhesion molecule-1 (ICAM-1) [51]. Additionally, zinc can indirectly modulate the NF-κB pathway by affecting the activity of immune cells that produce cytokines and other molecules involved in NF-κB signalling [52]. Furthermore, zinc can play a role in epigenetic regulation by influencing DNA methylation and histone modifications, potentially affecting the expression of genes involved in the NF-κB signalling pathway [53].

Line 206/210/220: Are the correct references [50], [51, 52], and [20] cited? Do these fit to the
text?

•    Response
Lines 249, 253. Thank you. The two misplacements have now been corrected (previous lines 206/210; reference in line 263, previous 220, is correct). 

Figure 1: In the Figure “Inflammaotory gene expression” should be changes to “Inflammatory
gene expression”. How das mtDNA affect TNF? The A20 circle in the left upper part should be in front of the line. “Nucleus” might be better located below the DNA. The trancription factors have to bind to the DNA to induce transcription of target genes, don´t them? Possibly a further figure showing the effect of zinc on transcription factors apart from NF-kappaB would be nice!

•    Response
Specified. Line 266. The figure has been corrected as per the reviewers’ suggestions. 

Lines 289-301. Regarding the question of how mtDNA affects TNF, this has now been clarified in the figure and explained and referenced in the text, expanding previous references on mtDNA, as follows: In an animal model, the presence of extracellular mitochondrial DNA (mtDNA) activated the NLR Family Pyrin Domain-Containing 3 (NLRP3) inflammasome, initiating inflammation via the Toll-like receptor 9 (TLR9), Mitogen-Activated Protein Kinase (MAPK), and Nuclear Factor-κB (NF-κB) pathways [67]. When monocytes were stimulated with mtDNA in vitro, it led to the production of TNF-α. Moreover, elevated levels of mtDNA in the plasma were correlated with increased in vivo levels of pro-inflammatory cytokines, including TNF-α, IL-6, Regulated upon Activation, Normal T Cell Expressed and Presumably Secreted (RANTES), and IL-1 receptor antagonist (IL-1ra) [68]. In both in vitro and in vivo settings, mtDNA activated neutrophils by promoting the phosphorylation of MAPK and inducing migration and degranulation of human polymorphonuclear neutrophils. This activation of neutrophils was mediated through Toll-like receptor-9 (TLR-9) and formyl peptide receptor-1 (FPR1) [69].

Lines 395. Finally, we drew a new figure showing the effect of zinc on some transcription factors apart from NF-κB. Accordingly, the following explanatory text was added to clarify critical zinc effects on other transcriptional factors. 

Lines 351-391. Stimulation of Nucleotide-binding oligomerisation domain-2 (NOD2) leads to an increase in intracellular zinc levels, which, in turn, enhances autophagy and promotes the clearance of bacteria. The regulation of metallothionein genes in human mono-cyte-derived macrophages (MDMs) relies on the zinc-dependent transcription factor MTF-1 [95]. Knocking down MTF-1 doesn't impact the baseline bacterial clearance by MDMs. However, when there is continuous NOD2 stimulation, the increase in intracellular zinc, autophagy, and bacterial clearance is compromised in MDMs lacking MTF-1 [95]. MTF-1 is involved in the transcription of genes responsible for sequestering and transporting zinc within cells and regulating intracellular signalling pathways by activating metallothioneins [96]. MTF-1 moves between the cytoplasm and nucleus, with its export likely mediated by its interaction with chromosomal maintenance 1 (Crm1), a major mammalian export protein facilitating the transport of large macromolecules, including RNA and proteins, across the nuclear membrane to the cytoplasm (Figure 2). The addition of zinc or induction of autophagy restores bacterial clearance in MDMs after metallothionein knockdown. In addition, in MDMs, NOD2 works synergistically with the Pattern Recognition Receptors (PRRs) TLRs 5 and 9 to enhance the effects of metallothioneins [95]. In mice, the intestinal microbiota plays a role in regulating the expression of metallothioneins, zinc levels, autophagy, and bacterial clearance in intestinal macrophages [95].
Similarly, in interaction with the activating transcription factor 4 (ATF4), HIF-1 improves the immune activity of macrophages involved in sepsis [97]. It has been demonstrated that inhibiting glycolysis contributes to neutrophil immunosuppression during sepsis, regulated by the PI3K/Akt-HIF-1α pathway-mediated downregulation of lactate dehydrogenase A (LDHA) [98]. SARS-CoV-2 ORF3a induces mitochondrial damage and the production of mitochondrial ROS, leading to increased HIF-1α expression, which subsequently facilitates SARS-CoV-2 infection and cytokine production [99]. The effects of HIF-1α on vasculature, metabolism, inflammation, immune response, and apoptosis contribute to lung injury in COVID-19 [100]. Zinc finger and Bric-a-brac (BTB) domain-containing protein 2 (ZBTB2) enhance the expression of specific HIF-1 target genes under hypoxia. HIF-1 binding to the consensus hypoxia-responsive element (HRE) sequence recruits ZBTB2 to the gene locus, increasing p300-mediated histone acetylation and enhancing gene expression under hypoxic conditions (Figure 2). In normoxia, prolyl hydroxylases (PHDs) hydroxylate HIF-1α on two proline residues in the oxy-gen-dependent degradation domain, triggering von Hippel–Lindau (pVHL)-mediated ubiquitination and proteasomal degradation. Concurrently, factor inhibiting HIF (FIH), an asparaginyl hydroxylase regulated similarly to PHDs in an oxygen-dependent manner, suppresses HIF-1's transcriptional activity in normoxia by preventing co-activator recruitment. In contrast, hypoxia inhibits PHDs and stabilizes HIF-1α, allowing it to translocate into the nucleus and dimerize with constitutively expressed HIF-1β, forming an active HIF-1 complex that triggers the transcription of genes promoting glycolytic metabolism, angiogenesis, and cell survival [101]. Both in vitro and in vivo studies confirm that zinc promotes angiogenesis through the HIF-1α/VEGF signal-ling pathway [102] (Figure 2)

Lines 245-248: How does zinc mediates these effects? This should be specified!

•    Response
Specified. Line 315-329. Zinc deficiency, both in vivo and in vitro, has been shown to result in impaired granulocyte migration and chemotaxis in response to N-formyl-L-methionyl-L-leucyl-L-phenylalanine (fMLF), as well as a reduction in the chemoattractant properties of IL-8 for T cells [73]. Moreover, zinc deficiency affects the modulation of the inflammatory response by IL-1ra on the cell surface [75]. In individuals with zinc deficiency, there is a demonstrated decrease in the exocytosis of neutrophil granules and secretory vesicles of macrophage-1 antigen (Mac-1), which is a complement receptor-3 (CR3), along with CD66b, a signal transducer involved in the adhesive activity of CD11/CD18 [73]. This reduction in exocytosis negatively impacts critical immune functions such as phagocytosis, oxidative burst, and granule release [76]. This reduction in exocytosis negatively impacts critical immune functions such as phagocytosis, oxidative burst, and granule release [77]. Furthermore, zinc deficiency leads to a decrease in the production of thymulin, which is essential for the differentiation and functioning of T cells [78]. It also impairs the production of Th1 cytokines, including TNF-α, IL-2, and IFN-γ, both in cell culture models [79] and in vivo [80].

Part 4 “Regulated cell death” should be condensed to avoid a repetition in parts 4.1 and 4.2.

•    Response
It was condensed. (most sentences were deleted, and a few were moved to more appropriate parts). Lines 286-290; 302-307. 

Line 363: Reference(s)?

•    Response
Added. Line 496.

Line 371: Between “[94]” and “Zinc” a full stop is missing.

•    Response
Added. Line 504 (now reference 131).

Line 375: “(eryptosis)” should be “Eryptosis”.

•    Response
Corrected. Line 508.

Lines 427-429: Is there a word missing?

•    Response
Added. Line 563.

Line 437: Necrostatin-1, Necrostatin-1s, or both? Please specify!

•    Response
Specified. Line 572.

Lines 476-479: Sentence doubled?

•    Response
Yes, thank you. Deleted. Line 609-611.

Line 571: “Numerous”? Reference [163] is only one study.

•    Response
Corrected. Line 707.

Line 576: “Acute Respiratory Distress Syndrome” should be “acute respiratory distress
syndrome”

•    Response
Corrected. Line 712

Part 6 “Plasma concentrations in critical illness, sepsis, COVID-19”: Is there a link to
mechanisms, which are affected by low or high zinc in these patients?

•    Response
Added. Line 740-757. The cytokine storm redistributes zinc from the plasma into cells as a protective measure to prevent immune dysfunction, systemic inflammation, oxidative stress that can lead to DNA damage, and cell apoptosis [210]. Additionally, the observed low zinc levels may result from a combination of a pre-existing deficiency, altered protein binding, increased losses, heightened metabolic demand, and dilution secondary to fluid overload [184]. These low zinc levels also contribute to immune dysfunction, which, in turn, increases ACE-2 activity, potentially facilitating binding with SARS-CoV-2. In vitro, it has been observed that the spike protein of the SARS-CoV-2 virus interacts with angiotensin-converting enzyme 2 (ACE2) and the serine protease transmembrane protease serine 2 (TMPRSS2) in the alveoli, allowing the virus to enter host cells. ACE2 is an enzyme that relies on zinc for its function and consists of two subdomains, I and II, specifically involved in zinc binding. Prior research has indicated that a decrease in zinc levels promotes the interaction between ACE2 and the spike protein of SARS-CoV-2 (Figure 4). The RNA synthesis of SARS-CoV-2 is catalyzed by an RNA-dependent RNA polymerase (RdRp), a core enzyme within its multiprotein replication and transcription complex [184]. During the acute phase of COVID-19, the inhibitory effect of zinc on virus entry, fusion, RNA synthesis, replication, protein translation, and viral dissemination is compromised due to insufficiently low zinc levels [231].  
Line 744: Zinc is not “expressed”.

•    Response
Corrected. Line 925.

Line 746: “hided” should be “hid” or “hidden.”

•    Response
Corrected. Line 928.

Some parts of the manuscript do not have a link to zinc (lines 278-328, 484-493). These should
be combined (see Part 4) including aspects on zinc.

•    Response
Some parts were deleted, and others were combined with more appropriate parts and rewritten including aspects on zinc. Lines 455-461; 520-524; 617-630.
